# Learning to Follow Instructions in Text-Based Games

**Mathieu Tuli, Andrew C. Li, Pashootan Vaezipoor, Toryn Q. Klassen[†],**
**Scott Sanner, Sheila A. McIlraith[†]**
University of Toronto, Toronto, Canada
Vector Institute for Artificial Intelligence, Toronto, Canada
[†] Schwartz Reisman Institute for Technology and Society, Toronto, Canada
{mathieutuli,andrewli,pashootan,toryn,sheila}@cs.toronto.edu
ssanner@mie.utoronto.ca

## Abstract

Text-based games present a unique class of sequential decision making problem in which agents interact with a partially observable, simulated environment via actions and observations conveyed through natural language. Such observations typically include instructions that, in a reinforcement learning (RL) setting, can directly or indirectly guide a player towards completing reward-worthy tasks. In this work, we study the ability of RL agents to follow such instructions. We conduct experiments that show that the performance of state-of-the-art text-based game agents is largely unaffected by the presence or absence of such instructions, and that these agents are typically unable to execute tasks to completion. To further study and address the task of instruction following, we equip RL agents with an internal structured representation of natural language instructions in the form of Linear Temporal Logic (LTL), a formal language that is increasingly used for temporally extended reward specification in RL. Our framework both supports and highlights the benefit of understanding the temporal semantics of instructions and in measuring progress towards achievement of such a temporally extended behaviour. Experiments with 500+ games in TextWorld demonstrate the superior performance of our approach.

## 1 Introduction

Building AI agents that can understand natural language is an important and longstanding problem in AI. In recent years, instrumented text-based game (TBG) engines have served as compelling environments for studying a variety of tasks related to language understanding, affordance extraction, memory, and sequential decision making (e.g., Côté et al., 2018; Adhikari et al., 2020; Liu et al., 2022). They provide a simulated, partially observable environment where an agent can navigate and interact with environment objects, receiving observations and administering commands via natural language. TextWorld (Côté et al., 2018) is a TBG learning environment for training reinforcement learning (RL) agents. Successful play requires language understanding, effective navigation, memory, and an ability to follow instructions embedded within the text. Instructions may or may not be directly bound to reward but can guide an RL agent towards completing tasks and collecting reward.

In this paper we study instruction following in text-based games and propose an approach that advances the previous state of the art. To this end, we employ the state-of-the-art model-free TBG RL agent called GATA (Graph Aided Transformer Agent) (Adhikari et al., 2020) that operates in the TextWorld environment. GATA has made significant advances in performance by augmenting TBG agents with long-term memory – a critical component of effective game play. Despite GATA's improvement over previous baselines, our experiments (see Figure 1) show that GATA performance is largely unaffected by the presence or absence of instructions, leading us to conclude that GATA is not effectively following instructions. We also find that while GATA agents are able to garner reward,

36th Conference on Neural Information Processing Systems (NeurIPS 2022).

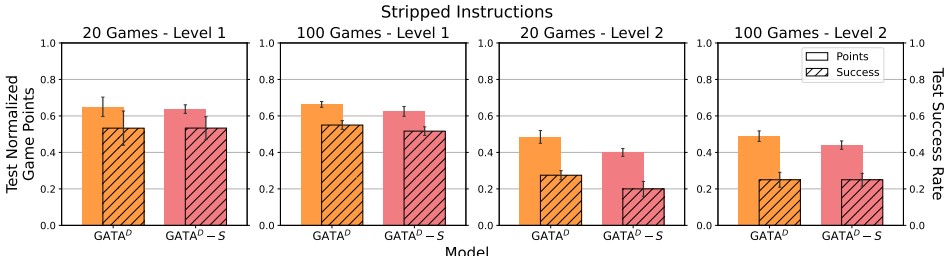

Figure 1: Comparison of GATA performance when trained with instructions (GATA[D]) versus when instructions are stripped from environment observations (GATA[D]-S). Agents were trained with 20 or 100 games, at increasing levels of task difficulty (level 1 vs level 2). Note that normalized game point performance (solid blocks) and rate of success (hashed blocks) are largely unchanged whether instructions are present or absent. Low success rate (i.e., task completion) rate is also seen in level 2.

they are not typically successful in *completing* tasks – an important vulnerability to the deployment of such techniques in environments where partial completion of tasks can be unsafe.

To further study and address the task of instruction following, we equip GATA with an internal structured representation of natural language instructions specified in Linear Temporal Logic (LTL) (Pnueli, 1977), a formal language that is increasingly used for temporally extended goal and preference specification in symbolic planning, and for reward specification, and other purposes in RL (e.g., Bacchus & Kabanza, 2000; Baier & McIlraith, 2006; Baier et al., 2009; Patrizi et al., 2011; Camacho & McIlraith, 2019; Littman et al., 2017; Toro Icarte et al., 2018a,b; Camacho et al., 2019; Leon et al., 2020; Kuo et al., 2020; Vaezipoor et al., 2021). LTL also provides a mechanism to monitor progress towards completion of instructions. Our framework both supports and highlights the benefit of understanding the temporal semantics of instructions and in measuring progress towards achievement of a temporally extended behaviour. We perform experiments that illustrate the superior performance of our TBG agent and its ability to follow instructions. Contributions of this work include:

- Experiments that expose the lack of instruction following and low task completion rate in a state-of-the-art TBG agent.

- An approach to the study and deployment of instruction following in TBG environments via exploitation of a formal language: LTL. LTL provides well-defined semantics and supports a measure of progress towards satisfaction of instructions.

- An augmentation to an existing state-of-the-art architecture for TBGs to equip a TBG agent with instruction-following capabilities.

- Comprehensive experiments and insights that study our and others' approaches to instruction following, and that highlight the superior performance of our proposed approach.

## 2    Background

In this section we introduce TextWorld, the TBG engine that we use, together with the Cooking domain that we employ in our experiments. We also overview Linear Temporal Logic, which (as described in section 1) we use in our approach as an internal representation for instructions.

### 2.1    Text-Based Games: TextWorld

Text-based games are partially observable multi-turn games where the environment and the player's action choices are represented textually. In this work, we use TextWorld (Côté et al., 2018) as our text-based game engine. A text-based game can be viewed as a (discrete-time) partially observable Markov decision process (POMDP) $\langle S, T, A, O, \Omega, R, \gamma \rangle$ (Côté et al., 2018) where $S$ is the environment's state space, $A$ is the action space, $T(s_{t+1}|s_t, a_t)$ where $s_{t+1}, s_t \in S$ and $a_t \in A$ is the conditional transition probability between states $s_{t+1}$ and $s_t$ given action $a_t$, $O$ is the set of (partial) observations that the agent receives, $\Omega(o_t|s_t, a_{t-1})$ is the set of conditional observation probabilities, $R : S \times A \to \mathbb{R}$ is the reward function, and $\gamma \in [0, 1]$ is the discount factor. An agent's goal is to learn some

optimal policy $\pi^*(a|o)$ (or a policy that conditions on historical observations or on some internal memory) that maximizes the expected discounted return. In this work, we focus on the choice-based variant of games, similar to previous works (Adhikari et al., 2020; Narasimhan et al., 2015). The action space $A$ is a list of possible commands and at each time-step $t$ in the game, the agent must select action $a_t \in C_t$ from the current subset of permissible actions $C_t \subset A$.

### 2.1.1 Environment Setting

We focus on the TextWorld *Cooking domain*, popularized by Adhikari et al. (2020) and Microsoft's First TextWorld Problems: A Language and Reinforcement Learning Challenge (FTWP) (Trischler et al., 2019). The game tasks agents with gathering and preparing various cooking ingredients described by an in-game recipe that is to be found. Game points (rewards) are earned for each of (1) collecting a required ingredient, (2) performing a preparatory step (some cutting or cooking action) on an ingredient as required by the recipe, (3) preparing the meal once all of the ingredients have been prepared, and (4) eating the meal. The game's partial observations can contain instructions that guide the agent towards completion of tasks, but not all instructions correspond directly to rewards. The game first instructs the agent to examine a cookbook, which elicits a recipe to be followed. The act of examining the cookbook returns no reward, but following its recipe will return reward. See Appendix C for more details. Success is determined by whether the recipe is fully completed and eaten. Preparing ingredients can also involve collecting certain tools (e.g., a knife). The game may also involve navigation – the agent may need to navigate to the kitchen or to find certain ingredients.

## 2.2 Linear Temporal Logic (LTL)

Linear Temporal Logic (LTL) (Pnueli, 1977) is a formal language – a propositional logical language with temporal modalities – that can be used to describe properties of trajectories. We will use LTL to specify instructions. LTL formulas are constructed from propositional variables (e.g., `player-has-carrot`), connectives from propositional logic (e.g., $\neg$, $\wedge$), and two temporal operators: $\bigcirc$ (NEXT) and $\mathsf{U}$ (UNTIL). Formally, we define the syntax of LTL per Baier & Katoen (2008) as

$$\varphi ::= p \mid \neg\varphi \mid \varphi \wedge \psi \mid \bigcirc\varphi \mid \varphi\,\mathsf{U}\,\psi$$

where $p \in \mathcal{P}$ for some finite set of propositional symbols $\mathcal{P}$. Satisfaction of an LTL formula is determined by a sequence of truth assignments $\sigma = \langle \sigma_0, \sigma_1, \sigma_2, \ldots \rangle$ for $\mathcal{P}$, where $p \in \sigma_i$ iff proposition $p \in \mathcal{P}$ holds at time step $i$. Formally, $\sigma$ *satisfies* $\varphi$ at time $i \geq 0$, denoted as $\langle \sigma, i \rangle \models \varphi$, under the following conditions:

- $\langle \sigma, i \rangle \models p$ iff $p \in \sigma_i$, where $p \in \mathcal{P}$
- $\langle \sigma, i \rangle \models \neg\varphi$ iff $\langle \sigma, i \rangle \not\models \varphi$
- $\langle \sigma, i \rangle \models \bigcirc\varphi$ iff $\langle \sigma, i+1 \rangle \models \varphi$
- $\langle \sigma, i \rangle \models (\varphi \wedge \psi)$ iff $\langle \sigma, i \rangle \models \varphi$ and $\langle \sigma, i \rangle \models \psi$
- $\langle \sigma, i \rangle \models \varphi\,\mathsf{U}\,\psi$ iff there exists $j$ such that $i \leq j$ and $\langle \sigma, j \rangle \models \psi$, and $\langle \sigma, k \rangle \models \varphi$ for all $k \in [i, j)$

A sequence $\sigma$ is then said to *satisfy* $\varphi$ iff $\langle \sigma, 0 \rangle \models \varphi$.

Any LTL formula can be defined in terms of $p \in \mathcal{P}$, $\neg$ (*negation*), $\wedge$ (*and*), $\bigcirc$ (NEXT), and $\mathsf{U}$ (UNTIL). From these operators, we can also define the Boolean operators $\vee$ (*or*) and $\rightarrow$ (*implication*), and the temporal operators $\square$ (ALWAYS) and $\lozenge$ (EVENTUALLY), where $\langle \sigma, 0 \rangle \models \square\varphi$ if $\varphi$ always holds in $\sigma$, and $\langle \sigma, 0 \rangle \models \lozenge\varphi$ if $\varphi$ holds at some point in $\sigma$.

### 2.2.1 LTL Progression

LTL formulas can also be *progressed* along a sequence of truth assignments (Bacchus & Kabanza, 2000; Toro Icarte et al., 2018b). In other words, as an agent acts in the environment, resulting truth assignments can be used to update the formula to reflect what has been satisfied. The updated formula would now reflect the parts of the original formula that are remaining to be satisfied or whether the formula has been violated/satisfied. The progression operator $\text{prog}(\sigma_i, \varphi)$ is defined as follows.

**Definition 2.1.** For LTL formula $\varphi$, truth assignment $\sigma_i$ over $\mathcal{P}$, and $p \in \mathcal{P}$, $\text{prog}(\sigma_i, \varphi)$ is defined as

- $\text{prog}(\sigma_i, p) = \begin{cases} \text{true} & \text{if } p \in \sigma_i \\ \text{false} & \text{otherwise} \end{cases}$
- $\text{prog}(\sigma_i, \neg\varphi) = \neg\,\text{prog}(\sigma_i, \varphi)$
- $\text{prog}(\sigma_i, \text{NEXT}\,\varphi) = \varphi$
- $\text{prog}(\sigma_i, \varphi_1 \wedge \varphi_2) = \text{prog}(\sigma_1, \varphi_1) \wedge \text{prog}(\sigma_1, \varphi_2)$
- $\text{prog}(\sigma_i, \varphi_1\,\text{UNTIL}\,\varphi_2) = \text{prog}(\sigma_1, \varphi_2) \vee (\text{prog}(\sigma_1, \varphi_1) \wedge \varphi_1\,\text{UNTIL}\,\varphi_2)$

In the context of TextWorld, the progression operator can be applied at every step in the episode to update the LTL instruction fed to the agent. To do so, it's necessary to have a *labelling function* that can indicate when propositions are true as the agent acts during an episode (e.g., to detect that `player-has-carrot` is true when the player has the carrot). We discuss how this labelling occurs in section 4, and give an example of how progression works in Appendix D.

## 3 Following Instructions with GATA

In order to evaluate the effectiveness of state-of-the-art text-based game agents at following instructions, we conducted experiments on the Cooking domain using the state-of-the-art model-free RL agent for TextWorld, GATA (Adhikari et al., 2020). GATA uses a transformer variant of the popular LSTM-DQN (Narasimhan et al., 2015) combined with a dynamic belief graph that is updated during game-play. The aim is to use this belief graph as long-term memory to improve action selection by modelling the underlying game dynamics (Adhikari et al., 2020). Formally, given the POMDP, GATA attempts to learn some optimal policy $\pi^*(a|o, g)$ where $g$ is the belief graph.

While GATA's belief graph can capture goal relations (e.g. `apple-needs-cut`), it turns out that agents trained to condition on observations and the GATA belief graph alone largely ignore in-game instructions. We tested a GATA agent on levels 1 and 2 in the Cooking domain, after training on either the 20-game or 100-game training set, and found that in none of those settings was the cookbook examined more than 15% of the time (3/20 testing games). In short, *the GATA agent usually doesn't observe what the recipe is for the current game*, meaning it has no way of knowing what the actual goal of the game is (except – eventually – from the rewards it gets and when the episode ends).

We further investigate how GATA agents fail to follow instructions by training these agents using modified game observations that have their instructions stripped (specifically, instructions directing the agent to examine the cookbook, the recipe text within the cookbook, and instructions to grab a knife if attempting to cut an ingredient without first holding the knife were removed from observations). This has two effects: (1) the agent no longer receives text-based instructions about what the goal is or what it should do; and (2) GATA's belief state will no longer capture goal relations like 'needs'. The results of this experiment are in Figure 1, and demonstrate how GATA's performance remains largely unchanged. This suggests that GATA is (here at least) (a) *not exploiting text-based instructions that would lead it to success* and (b) *even not exploiting the goal-related relations in its own belief state*.

The results in Figure 1 also show a drop in GATA's performance when moving from level 1 to level 2 in the Cooking domain, where the games' complexity is increased by just one added ingredient preparation step in the recipes (see Table 1 for more details on the levels). GATA has difficulty in fully completing tasks on level 2 games, where its success rate is roughly half that of its achieved normalized game points (only the latter metric was used by Adhikari et al. (2020)).

Given these insights, we wish to further study and address instruction following in TBGs. In the next section, we propose using LTL and demonstrate how existing work can be easily augmented.

## 4 An Approach to Following Instructions

We now investigate a mechanism for both studying and advancing the ability of an RL agent to follow instructions. We do so by translating instructions to an internal structured representation of language in the form of LTL, a formal language that is increasingly being used for reward specification in RL agents (Vaezipoor et al., 2021; Leon et al., 2020; Kuo et al., 2020; Camacho et al., 2019; Toro Icarte et al., 2018b). We describe how to augment the GATA architecture with these LTL instructions and how to monitor progress towards their completion.

### 4.1 Generating and Representing LTL Instructions for TextWorld

We use three types of LTL instructions for the Cooking domain. The first instruction identifies the need to examine the cookbook, and is the formula `NEXT cookbook-is-examined`. This instruction simply states that the agent should examine the cookbook (i.e. `cookbook-is-examined = true`) in the next step of the game. The second instruction is the actual recipe that gets elicited from the cookbook. We format this instruction to be *order-invariant* and *incomplete*. Order-invariance allows the agent to complete the instructions in any order, but is still constrained by any ordering

that the TextWorld engine may enforce. "Incomplete" simply refers to the fact that not every single action required to complete the recipe is encoded (e.g., grabbing a knife before slicing a carrot, or opening the fridge). The agent must still learn to do these things to accomplish its tasks, but is not directly instructed to do so. Assuming the recipe requires that predicates $p_1, p_2, \ldots p_n$ be true, the cookbook instructions are modelled as $(\texttt{EVENTUALLY } p_1) \wedge (\texttt{EVENTUALLY } p_2) \wedge \ldots (\texttt{EVENTUALLY } p_n)$. For example, in the Cooking Domain, this instruction might be the conjunction

$$(\texttt{EVENTUALLY apple-in-player}) \wedge (\texttt{EVENTUALLY meal-in-player}) \wedge (\texttt{EVENTUALLY meal-is-consumed}).$$

The third and final type of instruction identifies the need to navigate to the kitchen. This instruction is defined as `EVENTUALLY player-at-kitchen`. This instruction will come prior to the first two described above, but is only used in games with navigation (see Table 1).

We build a simple LTL translator that generates these instructions from the textual observations, similar to the goal generator used in Liu et al. (2022). TextWorld's observations are easily parsed to extract the goal information already contained within them, which we then formalize and keep track of using LTL. We provide examples of these observations and more details in Appendix E. Note that these observations are only used to generate the instruction itself, and subsequently LTL progression is used, with the GATA belief state as our labelling function, to monitor completion of instruction steps and to update instructions that remain to be addressed.

One possible criticism with such an LTL translator is its reliance on domain knowledge. While not the main focus of this paper, a complementary research problem that has begun to be explored is to *automatically* translate natural language instructions to LTL (e.g., Dzifcak et al., 2009; Finucane et al., 2010; Wang et al., 2020). Traditionally, such approaches have required large corpora of training data or hard-coded rules, and were restricted to a specific domain. However, pretrained large language models such as GPT-3 introduce the potential for a general natural-language-to-LTL translation scheme with minimal domain-specific adaptation (Hahn et al., 2022; Huang et al., 2022; Brohan et al., 2022). We explore this prospect by applying GPT-3 to TextWorld in subsection 5.5.

Finally, we note that in this work, GATA provides the domain-dependent vocabulary for describing properties of state (e.g. `carrot-is-chopped`) while our LTL augmentation provides the *domain-independent* temporal modalities (i.e., `NEXT`, `EVENTUALLY`, etc.) and the logical connectives for composing those properties of state into the instructions we use. In this way, our technique is very generalizable, limited only by the recognizable properties of state (which in our case are provided by GATA) and instructions that can be extracted in game-play.

## 4.2   LTL Augmented Rewards and Episode Termination

We can also define an augmented reward function $R_{\text{LTL}}(s, a, \varphi)$, where $\varphi$ is an LTL formula, that rewards the agent for completing instructions. Given a labelling function $L : S \times A \to 2^{\mathcal{P}}$ that assigns truth values to the propositions in $\mathcal{P}$,

$$R_{\text{LTL}}(s, a, \varphi) = R(s, a) + \begin{cases} 1 \text{ if } \text{prog}(L(s, a), \varphi) = \textsf{true} \\ -1 \text{ if } \text{prog}(L(s, a), \varphi) = \textsf{false} \\ 0 \text{ otherwise} \end{cases}$$

In other words, a bonus reward is given for every LTL instruction the agent satisfies and a penalty is given if the agent fails to complete an instruction. We perform an ablative study on the effect of this reward in Appendix H.3.2. We henceforth refer to this modified reward function as the *LTL reward*. The maximum bonus reward an agent receives is either 2 if there is no navigation task, or 3.

Further, because we wish to *satisfy* instructions, we can also use the instructions to modify episode termination. That is, if our LTL instruction is violated, we have arrived in a terminal state, even if TextWorld has not indicated so. We perform an ablative study on the effect of this *LTL-based termination* in Appendix H.3.2.

## 4.3   LTL-GATA Model Architecture

We build a similar model to GATA's original architecture, augmented to include the LTL encoding of instructions and their progression according to observed system state. We dub this model LTL-GATA, which we describe in detail below. Figure 2 depicts an episode step interaction of LTL-GATA with TextWorld and Figure 3 depicts the model itself. Additional details can be found in Appendix F.

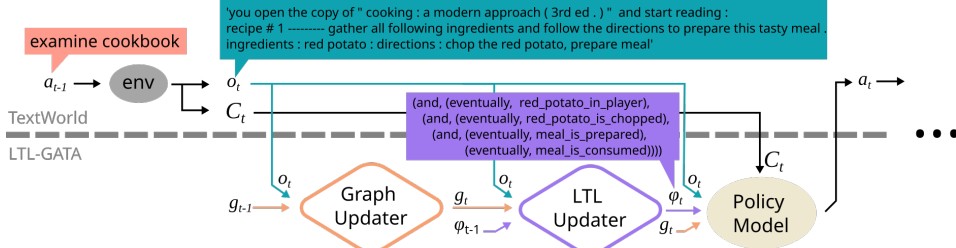

Figure 2: An example of a single step in an episode of TextWorld. The game environment returns an observation $o_t$ and action candidate set $C_t$ in response to action $a_{t-1}$. In turn, the agent's graph updater (GATA) updates its belief graph $g_t$ in response to both $o_t$ and $g_{t-1}$. Next, $g_t$ and $o_t$ update the LTL instructions. $\varphi_t$ is generated from $o_t$ after the cookbook is examined and thereafter $\varphi_{t-1}$ is progressed to $\varphi_t$ at each time step. The policy network selects action $a_t$ from $C_t$ conditioned on $o_t$, $\varphi_t$, and $g_t$ and the cycle repeats.

**Graph Updater:** We use the original GATA-GTP model (Adhikari et al., 2020), which generates a discrete belief graph as a list of triplets of the form (*object*, *relationship*, *object*). It is composed of two sub-components: (a) the belief state updater, which generates $g_t$ from observation $o_t$ and the graph $g_{t-1}$; and (b) the graph encoder, which encodes the current graph into a vector as $\mathtt{GE}(g_t) = g'_t \in \mathbb{R}^D$ for some latent dimension $D$. The graph encoder is a relational graph convolutional network (R-GCN) (Schlichtkrull et al., 2018) using basis regularization (Schlichtkrull et al., 2018) and highway connections (Srivastava et al., 2015). We refer the readers to Adhikari et al. (2020) for more details.

**LTL Updater:** The LTL updater generates and progresses LTL instructions. LTL instructions defining the need to arrive at the kitchen and examine the cookbook are generated from the initial observation $o_0$. The subsequent instruction defining the recipe is generated from game observation $o_t$, as described in subsection 4.1, when the action *examine cookbook* is executed at time $t$. For the truth assignments (i.e. the labelling function $L$), we leverage GATA's highly accurate belief state from the graph updater. We use the Spot engine (Duret-Lutz et al., 2016) to perform the progression.

**Text Encoders:** For encoding the action choices $C_t$, observations $o_t$, as well as encoding the LTL instructions $\varphi_t$, we use a simplified version of the Transformer architecture presented by Vaswani et al. (2017). This is the same architecture used by Adhikari et al. (2020). For LTL instructions, we encode them directly as a string. For example, the LTL formula $\varphi = (\mathtt{EVENTUALLY}\, p_1) \wedge (\mathtt{EVENTUALLY}\, p_2)$, where $p_1 = \mathtt{pepper\text{-}in\text{-}player}$ and $p_2 = \mathtt{pepper\text{-}is\text{-}cut}$, has the string representation

$$\mathrm{str}(\varphi) : \text{"eventually player\_has\_pepper and eventually pepper\_is\_cut"}$$

We format each predicate as a single token, and we show in Appendix H.3.1 that our method is robust to predicate format. For some input string $v \in \mathbb{R}^\ell$ of length $\ell$, the text encoder outputs a single vector $\mathtt{TE}(v) = v' \in \mathbb{R}^D$ of dimension $D$, which is the same latent dimension as the graph encoder.

**Action Selector:** The action selector is a 2-layer multi-layer perceptron (MLP). The encoded state vectors $\mathtt{TE}(o_t) = o'_t \in \mathbb{R}^D$, $\mathtt{TE}(\varphi_t) = \varphi'_t \in \mathbb{R}^D$, and $\mathtt{GE}(g_t) = g'_t \in \mathbb{R}^D$ are concatenated to form the agent's final state representation $z_t = [o'_t; \varphi'_t; g'_t] \in \mathbb{R}^{3D}$. In contrast to Adhikari et al. (2020), we concatenate features rather than use the bi-directional attention-based aggregator. This simplified the model's complexity and worked just as well experimentally. This vector is then repeated $n_c$ times and concatenated with the encoded actino choices $C'_t \in \mathbb{R}^{n_c \times D}$ where $n_c$ is the number of action choices. This input matrix is fed to the MLP which returns the a vector of Q-values for each action $q_c \in \mathbb{R}^{n_c}$.

**Training.** Formally, for belief state $g$ and LTL instruction $\varphi$, LTL-GATA aims to learn an optimal policy $\pi^*(a|o, g, \varphi)$. To learn this optimal policy, we implement Double DQN (DDQN) (Van Hasselt et al., 2016) with reward function and termination criteria as discussed in subsection 4.2. We use a prioritized experience replay buffer (Schaul et al., 2016). Refer to Appendix G.2 for further details.

## 5 Experiments

Our experimental assessment was designed both to understand how well GATA was exploiting observational instructions, as discussed in section 3, and to assess the instruction-following performance

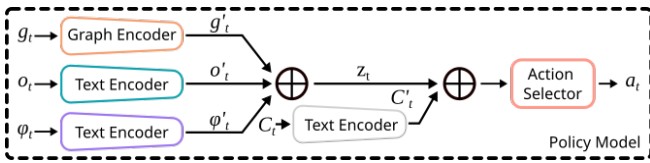

Figure 3: LTL-GATA's policy model. The model chooses action $a_t \in C_t$ conditioned on the state $z_t = [o'_t; \varphi'_t; g'_t]$. The action selector chooses $a_t$ based on the predicted Q-values.

Table 1: Cooking Levels

| Level | Recipe Size | Rooms | Max Score | Need (Grab, Cut, Cook) |
|---|---|---|---|---|
| 0 | 1 | 1 | 3 | {✓, ✗, ✗} |
| 1 | 1 | 1 | 4 | {✓, ✓, ✗} |
| 2 | 1 | 1 | 5 | {✓, ✓, ✓} |
| 3 | 1 | 9 | 3 | {✓, ✗, ✗} |

of our proposed approach relative to this state of the art (not only in terms of game points but also successful completion). We additionally strove to assess features of our approach (such as monitoring instruction progress) that contributed to its performance, as well as general challenges to text-based game playing that limited its performance (such as navigation).[1]

### 5.1 Experimental Setup

**Games.** To have as fair a comparison with Adhikari et al. (2020) as possible, we reused the sets of games they had generated. For the training games, they had created two sets: one set that contains 20 unique games per level and another that contains 100 unique games per level. Both the validation and testing sets have 20 unique games each per level. The levels we chose to use in our assessment are shown in Table 1. Note that in our assessment we omit Levels 4 and 5. Level 4 is an augmentation to Level 3 that adds more ingredients; both GATA and LTL-GATA at this level suffer from the navigation issues we discuss later with respect to Level 3. As we wanted to focus on instruction following and not navigation, we omitted this level and chose to use Level 0 instead. Level 5 is simply a random combination of all levels, so it is omitted for similar reasons.

**Hyper-parameters.** We replicate all but three hyper-parameters from Adhikari et al. (2020): (1) we use a batch size of 200 instead of 64 when training on the 100 game set, (2) for level 3, we use Boltzmann action selection, and (3) we use Adam Kingma & Ba (2015) with a learning rate of $0.0003$ instead of RAdam Liu et al. (2020) with a learning rate of $0.001$. These changes boosted performance for all models. See Appendix H.1 for more details.

**Baselines.** We compare against (1) TDQN (Adhikari et al., 2020), the transformer variant of the LSTM-DQN (Narasimhan et al., 2015) model, (2) GATA$^C$, and (3) GATA$^D$. GATA$^C$ is GATA's best performing model (GATA-COC) that uses a continue graph-updater pre-trained using contrastive observation classification. GATA$^D$ is a similarly performant model (GATA-GTP) that uses a discrete graph-updater pre-trained with ground-truth graphs from the FTWP dataset. Finally, we note that we found a few issues with GATA's original code[2] and have since fixed them (see Appendix H.4.1). For comparison, we include the original *paper* GATA models, labelled as GATA$_P^C$ and GATA$_P^D$.

**Measuring performance.** We measure performance using two metrics: normalized accumulated game points and game success rate. We report averaged results over 3 seeds for each experiment. Previous works only compared using the normalized accumulated game points; however, this may sometimes be misleading — an agent could get $3/4 = 0.75$ points on all games but never actually succeed on any. In contrast, measuring the success rate alongside the normalized game points allows for a more complete analysis of the agent's ability to play and complete these games.

### 5.2 LTL-GATA Compared to Baselines

**Consistently high performance with 20 training games**. We see from Figure 4 that LTL-GATA exhibits consistently high performance across levels as compared to the baselines when trained on the 20 games set. In particular, LTL-GATA maintains its performance on level 2, where the game's slight increase in complexity causes large performance drop-offs in other methods. Our agent can easily complete the added task and maintain similar performance to the previous level 1.

---

[1]Our code for the experiments can be found at `https://github.com/MathieuTuli/LTL-GATA`

[2]https://github.com/xingdi-eric-yuan/GATA-public, released under the open-source MIT License.

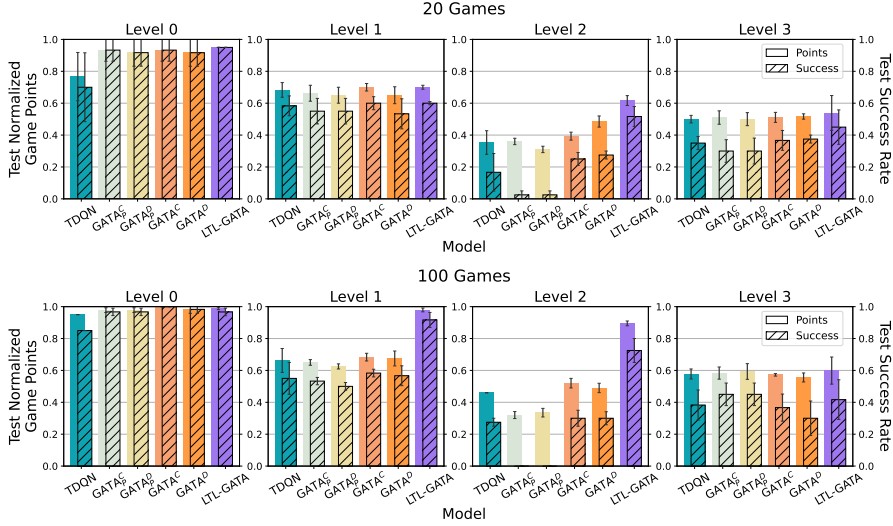

Figure 4: Testing scores across various levels and on both the 20 (top) and 100 (bottom) game training sets. We select the top-performing models (per seed) on the validation set during training and apply those models on the test set and report the average scores.

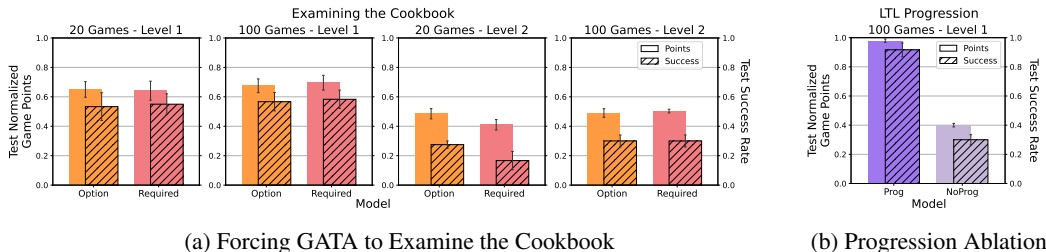

(a) Forcing GATA to Examine the Cookbook

(b) Progression Ablation

Figure 5: (a) A comparison of GATA$^D$ performance when given the *Option* to examine the cookbook vs. when it is *Required* to examine the cookbook. (b) A comparison of LTL-GATA with (*Prog*) and without (*NoProg*) using LTL progression.

**Large performance gains with 100 training games**. We see from Figure 4 that LTL-GATA gains considerable performance when trained on 100 games. With the added games, our agent is exposed to more predicates and can now generalize better to the testing set. Future work may look at how to achieve this kind of generalization without having to expose our agent to more predicates.

**Success rate and normalized game points.** Looking at the performance of GATA on level 2, it becomes apparent why measuring the success is important. Although it achieves almost $0.4$ normalized points, the actual success rate is near $0$ for original GATA models, and $\sim 60\%$ of the normalized points for the fixed models average across both training sets. In contrast, LTL-GATA exhibits high normalized points and success rate, where the average success rate across both training sets is $\sim 82\%$ of the normalized points.

**Competitive performance on level 3.** Level 3 introduces the added challenge of navigation. LTL-GATA outperforms GATA in this level as well, but not to the degree of previous levels. Inspecting testing trajectories, it becomes evident that both LTL-GATA and GATA methods struggle with navigation in this level, and have difficulties even navigating to the kitchen in the first place. Exploring at test time to find items and rooms in an unknown environment is a major challenge built into text-based games. Hypothetically, LTL could contribute to addressing this challenge. LTL could be used to dictate strategy and/or to simply track such exploration (e.g., for remembering which rooms have been previously visited). LTL might also be used to encode *learned* navigation instructions (e.g. "find the blue door, go through it, then turn right"). We do not pursue this vector of research here, but it is an interesting direction for future work.

## 5.3 Does LTL Progression Matter?

We show in Figure 5b that the use of progression is critical to performance, where LTL-GATA without progression incurs a large performance drop-off, dropping below the performance of the baselines as well. Without progression, the LTL instruction will not reflect the changes incurred by the agent's actions. This appears to confuse the agent considerably, demonstrated by its performance drop-off.

## 5.4 Forcing GATA to Examine the Cookbook

Because LTL-GATA is always tasked with examining the cookbook, we question whether a similar tasking for GATA improves performance. We experiment with GATA$^D$ by forcing the agent to examine the cookbook on the first step of the episode. Forcing GATA to examine the cookbook will elicit goal relations like (apple,needs,cut) in the belief state. We show however in Figure 5a that GATA does not improve when being given the cookbook. This shows that GATA cannot make use of the information elicited from the cookbook, continuing to ignore important instructions. Even with the presence of goal relations in its belief state, GATA fails to properly attend to this information. This highlights the benefits of a formalized representation of instructions used by LTL-GATA.

## 5.5 On Automatic Translation: Natural Language Instructions to LTL

While LTL-GATA relies on a handcrafted LTL translator to provide initial instructions from text observations, we investigate the potential of automating this step using pretrained large language models. This is not a central focus of the paper. Rather, we include this exploration as a proof of concept that the use of LTL is not a barrier to broad deployment of the work presented here. To this end, we evaluate whether GPT-3 (Brown et al., 2020) can few-shot learn to translate TextWorld observations to LTL, given only *six* examples and without additional training.

We experiment with two models of GPT-3 from OpenAI: *Ada* (the fastest model) and *Da Vinci* (the most powerful model). We perform few-shot translation by constructing prompts that contain six example translations, followed by the natural language observation to translate (the test case). The examples remain fixed for all test cases, and follow the form "NL: <natural language observation>. LTL: <ltl-formulas>". Our test case follows the form "NL: <natural language observation>. LTL:", where the model must complete the prompt, thereby performing a translation. We consider a response that exactly matches the ground-truth LTL formula as *absolutely correct*, a response that is otherwise correct except for parentheses and spaces as *almost correct*, and all other responses as *incorrect*. Further details and examples can be found in Appendix H.6.

Out of 234 test cases, *Da Vinci* translated 93.2% *absolutely correctly* and another 5.6% *almost correctly*, with only 1.3% of examples incorrect. *Da Vinci* displayed an impressive ability to generalize to unseen adjectives (e.g. is_grilled), nouns (e.g. carrot), and compositions of formula. Unfortunately, the weaker model, *Ada*, translated all 100% of examples incorrectly. We found that *Ada* commonly hallucinated new nonsensical words and predicates such as ingredient_is_salt_is_diced or banana_pork_chop_in_player, leading to erroneous translations.

## 6 Related Work

**Text-based games.** In this work we equip a text-based deep RL agent with formalized LTL instructions, building on previous works that employed belief graphs for solving text-based games. Adhikari et al. (2020) focused on supervised (i.e. translation) and self-supervised learned mechanisms to construct such belief graphs, whereas Ammanabrolu & Hausknecht (2020); Yin & May (2019b); Ammanabrolu & Riedl (2019) employed rule-based methods. At a larger scope, there is a host of other works on playing text-based games using deep reinforcement learning (Hausknecht et al., 2020; Zahavy et al., 2018; Jain et al., 2020; Yin & May, 2019a). Yuan et al. (2018) used count-based memory to shape the reward to improve in exploration and generalization in a simple domain. Narasimhan et al. (2015) and He et al. (2016) proposed variations of an LSTM-based model, which the TDQN model, used in this work, is built from. In just published work, Liu et al. (2022) took a model-based approach, focusing on object-oriented dynamics. However, these works do not address the role and representation of instructions that defines our work. Kimura et al. (2021) does employ a neuro-symbolic RL method using Logical Neural Networks. However, it does not focus on instructions, operates over all logical facts of the environment, and is applied to a simpler domain.

**Instruction following and Linear Temporal Logic.** Vaezipoor et al. (2021) trained an RL agent to follow various LTL instructions in both discrete and continuous action-space visual environments. They used R-GCNs to learn representations of the LTL instructions and also employed LTL progression. Their model showed good generalization performance on similar and much larger unseen instructions than those observed during training . However, in contrast to the work presented here, they relied on ground-truth labelling functions and operated in fully observable settings, while we use GATA's learned belief graphs, in a partially observable setting, to evaluate the truth or falsity of propositions and to progress formulae. We further distinguish ourselves from this work by opting for training the LTL semantics end-to-end using a transformer rather than an R-GCN. Works using LTL for reward specification (Leon et al., 2020; Kuo et al., 2020; Camacho et al., 2019; Toro Icarte et al., 2018b; Littman et al., 2017) or advice (Toro Icarte et al., 2018a) in RL agents exist, however they do not focus on text-based environments nor partially observable ones.

## 7 Conclusion

We studied the ability of RL agents to follow instructions in text-based games using TextWorld. We conducted experiments to show how current state-of-the-art model-free agents largely fail to exploit instructions and do not typically complete prescribed tasks. We then showed how LTL can be used to construct internal structured representations for state augmentation that result in large performance improvements and more reliable instruction following and task completion. Experiments showed that monitoring instruction progress was critical to these gains. Our method inherits limitations in dealing with navigation and unseen games from prior work, but these concerns are somewhat orthogonal to our focus on instruction following.

Furthermore, we can consider the broader impact of this work by relating to the critical need for good instruction following in safety-oriented domains such as autonomous transport or health care. We would like to suggest that works towards building better language agents should also emphasize the importance of *completing* instructions. To illustrate, for an agent to help a person half-way across a street, or to start but not finish a medical operation, may be worse than for it to do nothing at all. To that end, we have proposed using (game) success rate as a metric for future work, and demonstrated how LTL-GATA is very successful in the games it plays, relative to the state-of-the-art. Overall, we intend this paper to highlight the importance of studying instruction following in environments like TextWorld that act as a proxies to the general class of problems dealing with language understanding and human-machine interaction.

Finally, in follow-on work we would like to explore more complex text-based games such as the Jericho environment (Hausknecht et al., 2020). These games involve a number of distinct challenges, including exploration, navigation, puzzle solving, language understanding, and instruction following. In this vein, we'd like to see whether LTL can be exploited to capture (learned) domain-specific strategic advice, or memory to tackle both navigation and exploration challenges. We'd like to further explore seamless ways to exploit the merits of natural language together with the benefits afforded by the compositional syntax and semantics of formal languages such as LTL. To this end, further advancing our explorations translating natural language to LTL is of interest and import, for this and a diversity of other applications in and outside RL.

## Acknowledgements

We thank the NeurIPS reviewers for their constructive feedback, and also the reviewers from the *Wordplay: When Language Meets Games* workshop at NAACL 2022, where a preliminary version of this paper appeared (Tuli et al., 2022). We gratefully acknowledge funding from the Natural Sciences and Engineering Research Council of Canada (NSERC), the Canada CIFAR AI Chairs Program, and Microsoft Research. Resources used in preparing this research were provided, in part, by the Province of Ontario, the Government of Canada through CIFAR, and companies sponsoring the Vector Institute for Artificial Intelligence (www.vectorinstitute.ai/partners). Finally, we thank the Schwartz Reisman Institute for Technology and Society for providing a rich multi-disciplinary research environment.

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
