# Supplementary Material to
# Learning to Follow Instructions in Text-Based Games

**Mathieu Tuli, Andrew C. Li, Pashootan Vaezipoor, Toryn Q. Klassen**[†]**,**
**Scott Sanner, Sheila A. McIlraith**[†]
University of Toronto, Toronto, Canada
Vector Institute for Artificial Intelligence, Toronto, Canada
[†] Schwartz Reisman Institute for Technology and Society, Toronto, Canada

## Contents

36th Conference on Neural Information Processing Systems (NeurIPS 2022).

## A   Reinforcement Learning

Reinforcement Learning (RL) is the problem of training machine learning models to solve sequential decision making problems. By interacting with an environment, RL agents must learn optimal behaviours given the current state of their environment. If the environment is fully observable, we can frame it as a Markov Decision Process (MDP) modelled as $\langle S, A, T, R, \gamma \rangle$ where $S$ is the environment's state space, $A$ is the action space, $T(s_{t+1}|s_t, a_t)$ where $s_{t+1}, s_t \in S$ and $a_t \in A$ is the conditional transition probability between states $s_{t+1}$ and $s_t$ given action $a_t$, $R : S \times A \to \mathbb{R}$ is the reward function which is used to compute the reward $r_t = R(s_t, a_t)$, and $\gamma \in [0, 1]$ is the discount factor. The goal for an RL agent is to learn some optimal policy $\pi^*(a|s)$ that maximizes the expected discounted return $\mathbb{E}_\pi \left[ \sum_{k=0}^\infty \gamma^k r_{t+k} \big| s_t = s \right]$. A single game is an *episode*, and steps in an episode are indexed by $t$.

## B   Partially Observed Reinforcement Learning

In a partially observed environment, an agent does not have access to the full state space $S$. We can frame this environment as a Partially Observable MDP (POMDP) modelled by $\langle S, A, T, O, \Omega, R, \gamma \rangle$. In this new setting, $\langle S, A, T, R, \gamma \rangle$ remain unchanged, $O$ represents the set of (partial) observations that the agent receives and $\Omega(o_t|s_t, a_{t-1})$ is the set of conditional observation probabilities. An agent's goal is to learn some optimal policy $\pi^*(a|o)$ (or a policy that conditions on historical observations or on some internal memory) that maximizes the expected discounted return.

## C   TextWorld: Cooking Domain

We present two examples of observations with instructions in Table 2 and highlight where the instructions are and where the rewards come from.

## D   An example of LTL progression

To illustrate how progression works, the LTL instruction (`EVENTUALLY player-has-carrot`) $\wedge$ (`EVENTUALLY player-has-apple`) would be progressed to (`EVENTUALLY player-has-apple`) once the agent grabs the carrot during an episode. In other words, when the agent reaches a state where `player-has-carrot` is true, the LTL instruction is progressed to reflect that the agent no longer needs to get the carrot but must still grab the apple at some point.

## E   Generating LTL in TextWorld

We provide some examples of the LTL instructions used in this work in Table 3, Table 4, and Table 5. We build a simple translator that reads game observations and constructs these LTL instructions directly, but only once. Repeated observations will not result in the same LTL formula being generated. Once a formula has been generated, LTL progression is used with the agent's belief state to progress the instructions along the truth assignments: observations are not directly used in the progression, although they do indirectly affect the progression by affecting the belief state.

For levels 0, 1, and 2, the LTL instructions that an agent can receive throughout an episode are (a) the task to examine the cookbook and (b) the recipe-bound task. In other words, the set of un-progressed instructions $\Phi$ it can receive over the course of an episode (assuming the cookbook is examined) is as follows:

$$\Phi = \{ \texttt{NEXT cookbook-is-examined},$$
$$(\texttt{EVENTUALLY } p_1) \wedge (\texttt{EVENTUALLY } p_2) \wedge \dots (\texttt{EVENTUALLY } p_n) \}$$

where the recipe requires that predicates $p_1, p_2, \dots p_n$ be true. Note that we also consider eating the meal to be a part of recipe in this case, although it is not explicitly mentioned in the recipe. Further, we note that the "prepare meal" task is represented by the predicate `meal-in-player`, as this is the event that occurs when the meal is prepared in the game.

Table 2: TextWorld observations for the Cooking Domain game. We show the observations and highlight where the instructions are, and finally identify what the rewards would be. This is for a level 2 game, and the total possible reward is 5.

| **Initial Game Observation** |
|---|
| ''You are hungry! Let's cook a delicious meal. Check the cookbook in the kitchen for the recipe. Once done, enjoy your meal!'' -= kitchen =- you find yourself in a kitchen. You start to take note of what's in the room. You can make out a closed fridge nearby. You can see an oven. You can make out a table. You wonder idly who left that here. You see a knife on the table. Something scurries by right in the corner of your eye. Probably nothing. You see a counter. The counter is vast. On the counter you see a raw red potato and a cookbook. You see a stove, but the thing is empty, unfortunately.'' |
| **Reward** |
| There is a reward of 1 given for eating the meal. i.e. the instruction ''Once done, enjoy your meal!'' will result in a reward of 1 *after* the recipe has been completed. Note that the instruction ''Check the cookbook in the kitchen for the recipe.'' is not bound to a reward. |

| **Observation following the examine cookbook action** |
|---|
| ''You open the copy of *''Cooking : a modern approach (3rd ed.)''* and start reading: recipe #1 ----- Gather all following ingredients and follow the directions to prepare this tasty meal. Ingredients: red potato: directions: chop the red potato, fry the red potato, prepare meal'' |
| **Reward** |
| There are 4 rewards from the instruction ''Gather all following ingredients and follow the directions to prepare this tasty meal. Ingredients: red potato: directions: chop the red potato, fry the red potato, prepare meal'': |

  • 1 for grabbing the red potato

  • 1 for chopping the red potato

  • 1 for frying the red potato

  • 1 for preparing the meal

For levels with navigation (i.e. level 3),

$$\Phi = \{\, \texttt{EVENTUALLY player-at-kitchen},$$
$$\texttt{NEXT cookbook-is-examined},$$
$$(\texttt{EVENTUALLY } p_1) \wedge (\texttt{EVENTUALLY } p_2) \wedge \ldots (\texttt{EVENTUALLY } p_n)\,\}$$

where the agent has the added task of first navigating to the kitchen. This instruction provides no help for actually how to arrive at the kitchen, only that the agent must do so. As a result, LTL-GATA still suffers from the difficulties of exploration, and perhaps investigating how LTL can be used to improve in navigation could be a direction for future work.

In total, LTL generation occurs only twice for any level, either during the initial observation or when the cookbook is read. When multiple instructions are generated at once, the agent will process them sequentially, in the order they are given.

## F  Model

### F.1  Text Encoder

The text encoder is a simple transformer-based model, with a transformer block (Vaswani et al., 2017) and word embedding layer. We use the pre-trained 300-dimensional fastText (Mikolov et al., 2017)

Table 3: Level 3 observation and resulting generated LTL instruction

| Observation |
|---|
| ''You are hungry!  Let's cook a delicious meal.  Check the cookbook in the kitchen for the recipe.  Once done, enjoy your meal!''  -= corridor =- ''You've entered a corridor.  There is a closed screen door leading west. You don't like doors?  Why not try going north, that entranceway is not blocked by one.  You need an exit without a door?  You should try going south.'' |
| **Generated LTL** |
| This observation will generate two instructions: First, |

$$(\text{EVENTUALLY player-at-kitchen})$$

and second,

$$(\text{NEXT cookbook-is-examined})$$

Table 4: Level 1 observation and resulting generated LTL instruction

| Observation |
|---|
| ''You open the copy of *"Cooking :  a modern approach (3rd ed.)"*  and start reading:  recipe #1 ----- Gather all following ingredients and follow the directions to prepare this tasty meal.  Ingredients:  red potato: directions:  chop the red potato, prepare meal'' |
| **Generated LTL** |

$$(\text{EVENTUALLY red-potato-in-player}) \land (\text{EVENTUALLY red-potato-is-chopped}) \land$$
$$(\text{EVENTUALLY meal-in-player}) \land (\text{EVENTUALLY meal-is-consumed})$$

word embeddings, which are trained on Common Crawl (600B tokens). These word embeddings are frozen during training. Strings are tokenized by spaces.

The transformer block is composed of: **(1)** a stack of 5 convolutional layers, **(2)** a single-head self-attention layer, and **(3)** a 2-layer MLP with ReLU non-linear activation function in between. The convolutional layers each have 64 filters, with kernel sizes of 5 and are each followed by a Layer Norm (Ba et al., 2016). We also use standard positional encoding (Vaswani et al., 2017). The self-attention layer uses a hidden size $H$ of 64. The Text Encoder outputs a single feature vector $v \in \mathbb{R}^D$, where $D = 64$ in our experiments.

### F.2    Encoder Independence

Figure 3 in the main paper visualizes each component of our model. Specifically, our model has four encoders: **(1)** Graph Encoder, **(2)** Text Encoder for observations, **(3)** Text Encoder for LTL instructions, and **(4)** Text Encoder for action choices. We note here that each of these encoders are independent models, trained concurrently. This is in contrast to the original GATA model that used the same Text Encoder for both the actions and the observations. Because these Text Encoders are relatively small transformers, there is no issues with fitting this model in memory. As shown in Table 6, the model is still quite efficient, even more than the original GATA code. We found that using independent encoders resulted in better performance than using a single Text Encoder that would have been responsible for encoding the observations, LTL instructions, and action choices.

Table 5: Level 2 observation and resulting generated LTL instruction

| Observation |
| --- |
| "You open the copy of *"Cooking : a modern approach (3rd ed.)"* and start reading: recipe #1 ----- Gather all following ingredients and follow the directions to prepare this tasty meal. Ingredients: red potato: directions: chop the red potato, fry the red potato, prepare meal" |

**Generated LTL**

$$\left(\texttt{EVENTUALLY}\;\texttt{red-potato-in-player}\right) \wedge \left(\texttt{EVENTUALLY}\;\texttt{red-potato-is-chopped}\right) \wedge$$
$$\left(\texttt{EVENTUALLY}\;\texttt{red-potato-is-fried}\right) \wedge \left(\texttt{EVENTUALLY}\;\texttt{meal-in-player}\right) \wedge$$
$$\left(\texttt{EVENTUALLY}\;\texttt{meal-is-consumed}\right)$$

## F.3 Action Selector

The action selector is a simple two-layer MLP with a ReLU non-linear activation function in between. It takes as input, at time step $t$, the concatenated representation of the agent's state vector $z_t \in \mathbb{R}^{3D}$ and the action choices $C'_t \in \mathbb{R}^{n_c \times D}$. Recall that in our experiments $D = 64$. The first layer uses an input dimension of $4D$ and an output dimension of $D$. The second layer has an input dimension of $D$ and output dimension of $1$, which after squeezing the last dimension during the forward pass, the final output vector $q_c \in \mathbb{R}^{n_c}$ represents the q-values for each action choice.

The input to the action selector is constructed by repeating the agent's state representation, $z_t$, $n_c$ times and then concatenating with the encoded actions choices $C'_t$. We wanted to further explain why this occurs, as it may not be immediately clear. The action selector in this work is a parameter-tied Q-value predictor. That is, for some action $a_i \in C_t$, $i \in [1, \ldots, n_c]$ and agent state representation $z_t$, the predicted Q-value is $q_i = \texttt{AS}([a_i, z_t])$. Thus, the action selector (i.e. $\texttt{AS}(\cdot)$) predicts Q-values given action $a_i$ and agent state representation $z_t$. Thus, during a single episode step, given our encoded actions choices $C'_t \in \mathbb{R}^{n_c \times D}$, in order for the action selector to predict Q-values for each of these action choices, we repeat $z_t \in \mathbb{R}^{3D}$ $n_c$ times and stack it together, which results in a state matrix $Z_t \in \mathbb{R}^{n_c \times 3D}$. When we concatenate this matrix with our action choices we are left with the input to our action selector: $[C'_t; Z_t] \in \mathbb{R}^{n_c \times 4D}$. Looking at this matrix, each row in this input matrix is effectively the concatenation of action $a_i$ with agent state representation $z_t$, and so passing this matrix to our action selector performs the parameter-tied Q-value prediction $q_i = \texttt{AS}([a_i, z_t])$ for all action choices, and outputs a single vector of Q-values for each action $q_c \in \mathbb{R}^{n_c}$. We can then use these predicted Q-values to perform action selection using a greedy approach, an $\epsilon$-greedy approach, Boltzmann action selection, etc.

# G   Implementation Details

## G.1   Augmenting GATA's Pre-Training Dataset

We note here that although possible, the vocabulary and dataset used by Adhikari et al. (2020) did not allow for the knowledge triple $\{cookbook, is, examined\}$ to be extracted from observations. Without this triple being extracted and added to the agent's belief state, there would be no way for the agent to progress LTL instructions requiring the agent to examine the cookbook. In our pre-training of the GATA graph encoder, we augmented the dataset provided by Adhikari et al. (2020) to include the triplet $\{cookbook, is, examined\}$ when relevant (i.e. when the agent examines the cookbook). This was a simple process of adding this triple to the ground truth belief graphs in the dataset so that during pre-training, GATA could learn how to translate these triplets from relevant observations.

### G.2 Training

For training to learn our optimal policy we use the Double-DQN (DDQN) (Van Hasselt et al., 2016) framework. We use $\epsilon$-greedy for training, which first starts with a warm-up period, using a completely random policy (i.e. $\epsilon = 1.0$) for the first $1,000$ episodes. We then anneal $\epsilon$ from 1.0 to 0.1 over the next $3,000$ episodes after the initial warm-up (i.e. episodes $1,000$ to $4,000$). We use a prioritized experience replay buffer ($\alpha = 0.6$ and $\beta = 0.4$) with capacity $500,000$. For DDQN, the target network updates occur every $500$ episodes. We update network parameters every $50$ game steps, and we play $50$ games in parallel.

We train all agents for $100,000$ episodes using a discount factor of $0.9$, and we use $\{123, 321, 666\}$ as our random seeds. Each episode during training is limited to a maximum of $50$ steps, and during testing/validation this limit is increased to $100$ steps. We report results and save checkpoints every $1,000$ episodes. We also use a patience window $p$ that reloads from the previous best checkpoint during training when validation performance has decreased for $p$ episodes in a row. This is the same strategy used in Adhikari et al. (2020). For our experiments, we used $p = 3$.

For reporting testing results, each model is trained using the three seeds mentioned before, and fine-tuned on the validation set. That is, the checkpoint of the model that performs best on the validation set during training is saved, and each of these models (three, one for each seed) is applied to the test set. Reported test results are the average over these three models.

## H   Experiments

### H.1   Hyper-Parameters

To have as fair a comparison as possible, we replicate all but three hyper-parameters from the settings used in Adhikari et al. (2020). We do this to remove any bias towards more finely tuned experimental configurations and focus only on the LTL integration. Further, we re-run the GATA experiments to confirm their original results. The three changes we implemented were (1) we use a batch size of 200 instead of 64 when training on the 100 game set, (2) for level 3, we use Boltzmann action selection, and (3) we use Adam (Kingma & Ba, 2015) with a learning rate of $0.0003$ instead of RAdam (Liu et al., 2020) with a learning rate of $0.001$. These changes boosted performance for all models. For the 20 training game set, we use a batch size of $64$.

For Boltzmann action selection, we used a temperature of $\tau = 100$. We experimented with various temperatures ($\tau \in \{1, 10, 25, 50, 100, 200\}$) and found $\tau = 100$ to perform the best across models.

### H.2   Computational Requirements

We report the wall-clock times for our experiments in Table 6.

### H.3   Additional Results

#### H.3.1   Ablation: Formatting LTL Predicates

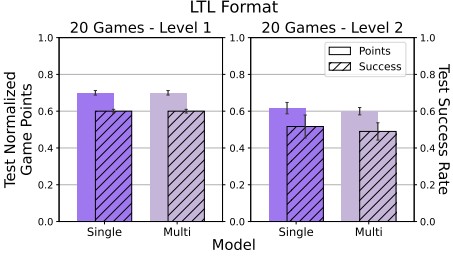

Figure 6: Study on LTL predicate format with single-token (*Single*) predicates and multi-token (*Multi*) predicates. Performance is largely unchanged with predicate format.

As we saw from Figure 4 in the main paper, LTL-GATA when trained on the 100 games set performs significantly better than when trained on the 20 game set, which we attribute to the increased exposure

Table 6: Training times for each model and training set size. The times were reported using a workstation with dual RTX3090s, an AMD Ryzen 5950x 16-core CPU, and 128GB of RAM. For the graph updater, COC stands for the contrastive observation classification pre-training (the continuous belief graph model) and GTP stands for ground-truth pre-training (the discrete belief graph model).

| Model | Training Set Size | Batch Size | Approximate Time |
|---|---|---|---|
| TDQN | 20 | 64 | 16 hours |
| LTL-GATA | 20 | 64 | 24 hours |
| GATA$^D$ | 20 | 64 | 24 hours |
| GATA$^C$ | 20 | 64 | 24 hours |
| GATA$^D_P$ | 20 | 64 | 36 hours |
| GATA$^C_P$ | 20 | 64 | 36 hours |
| TDQN | 100 | 200 | 32 hours |
| LTL-GATA | 100 | 200 | 48 hours |
| GATA$^{D*}$ | 100 | 200 | 48 hours |
| GATA$^C$ | 100 | 200 | 48 hours |
| GATA$^D_P$ | 100 | 200 | 65 hours |
| GATA$^C_P$ | 100 | 200 | 65 hours |
| Graph Updater using COC | N/A | 64 | 48 hours |
| Graph Updater using GTP | N/A | 64 | 48 hours |

to predicates during training, allowing it to generalize better during testing. To see if we can achieve the same level of generalization when training on the 20 game set, we compare LTL-GATA with LTL predicates represented as single tokens (what we did in the main paper) with using multiple tokens. That is, we compare the following two string representations:

(single-token predicates)  str$(\varphi)$ : "eventually player_has_pepper and eventually pepper_is_cut"

(multi-token predicates)  str$(\varphi)$ : "eventually player has pepper and eventually pepper is cut"

The single-token predicates are mapped in the vocabulary to a single word embedding. In our work, we compute word embedding for these single-token predicates by averaging the word embeddings of each underscore-separated word in the predicate. For example, the word embedding (`WE`) of the token `player_has_pepper` is

$$\mathtt{WE}(\mathtt{player\_has\_pepper}) = \frac{\mathtt{WE}(\mathtt{player}) + \mathtt{WE}(\mathtt{has}) + \mathtt{WE}(\mathtt{pepper})}{3}$$

For multiple-token predicates, each word has its own word embedding and we treat each word as any other word in the sentence. The idea is that by separating the tokens in the predicates, the text encoder (transformer) may be able to attend to each token independently, and during testing have better generalization. We visualize the results of this study in Figure 6. We can see from Figure 6 that this in fact does not help, and LTL-GATA performs almost equally in either scenario. This does however show how our method is robust to predicate format.

### H.3.2   The Effect of LTL Reward and LTL-Based Termination

It is important to study the effect that the additional LTL bonus reward and LTL-based episode termination has on the performance of LTL-GATA. To study this, we consider three scenarios: **(a)** LTL-GATA with the new reward function $R_{\text{LTL}}(s, a, \varphi)$ and without LTL-based episode termination; **(b)** LTL-GATA with the base TextWorld reward function $R(s, a)$ and LTL-based episode termination; and **(c)** LTL-GATA with the normal TextWorld reward function $R(s, a)$ and without LTL-based episode termination. For (a) and (b) we select level 1 on both the 20 and 100 game training set and level 2 on the 20 game training set. For (c) we select level 2 on 20 training games. We visualize the ablative study of these three scenarios in Figure 7.

From Figure 7 we can conclude that the presence of either the new reward function $R_{\text{LTL}}(s, a, \varphi)$ *or* LTL-based episode termination is important to the performance of LTL-GATA. This is because either of these methods will incentivize the agent to complete the initial `NEXT cookbook-is-examined`

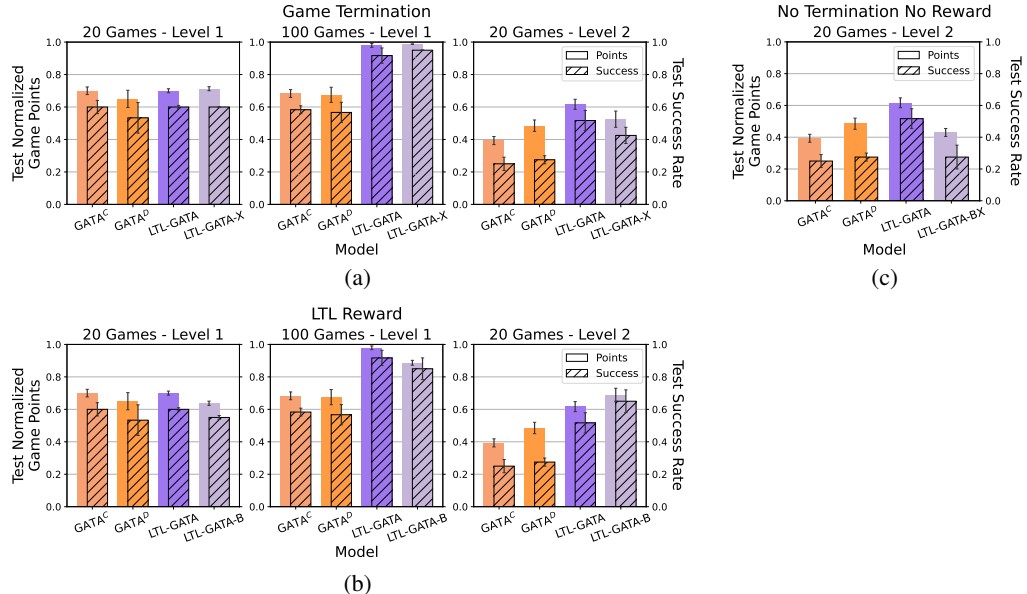

Figure 7: Ablation studies on LTL-based episode termination and new reward function $R_{\mathrm{LTL}}(s, a, \varphi)$ and on LTL progression. (a) LTL-GATA with new reward function $R_{\mathrm{LTL}}(s, a, \varphi)$ and without LTL-based episode termination (*LTL-GATA-X*). (b) LTL-GATA with base game reward function $R(s, a)$ and with LTL-based episode termination (*LTL-GATA-B*). (c) LTL-GATA with base game reward function $R(s, a)$ and without LTL-based episode termination (*LTL-GATA-BX*).

instruction, which isn't intrinsically rewarded by TextWorld. We can demonstrate the importance of this incentive by analyzing just one level (level 2 on 20 training games). Removing both methods leads to the agent not examining the cookbook, preventing it from receiving further instructions, which we can see from Figure 7(c) results in considerable performance loss, regressing to the baseline GATA.

## H.4 Code

All code for this work can be found at `https://github.com/MathieuTuli/LTL-GATA`.

### H.4.1 Fixing the GATA code

We found two primary issues in the GATA code. First, we noticed that their implementation of the double Q-learning error was wrong. For Double Q-Learning, after performing some action $a_t$ in state $s_t$ and observing the immediate reward $r_t$ and resulting state $s_{t+1}$, the Q-Learning error is defined per Van Hasselt et al. (2016) as

$$Y_t = r_t + \gamma Q(s_{t+1}, \arg\max_a Q(s_{t+1}, a; \boldsymbol{\theta}_t); \boldsymbol{\theta}'_t) \tag{1}$$

where $\boldsymbol{\theta}_t$ and $\boldsymbol{\theta}'_t$ are the parameters of the policy network and the target network, respectively. However, we noticed that the original code for GATA was computing the error as [1]

$$Y_t = r_t + r_{t+1} + \gamma Q(s_{t+1}, \arg\max_a Q(s_{t+1}, a; \boldsymbol{\theta}_t); \boldsymbol{\theta}'_t)$$

In other words, the reward for the stepped state was also being added to the error.

Second, we found that the double Q-learning error for terminal states was being incorrectly implemented. Specifically, when computing the error for the case where $s_t$ is a terminal state, and

---

[1] `https://github.com/xingdi-eric-yuan/GATA-public/blob/c1afc3c9ab38256f839b3e0ddf8243796df5bd77/dqn_memory_priortized_replay_buffer.py#L120-L123`

therefore the stepped state $s_{t+1}$ does not exist, the stepped state was not being masked [2]. Additionally, presumably because of this initial error, terminal states were very rarely returned when sampling from experience, unless certain criteria were met [3].

We found fixing these issues improved GATA's performance considerably, which we demonstrated in Figure 4 in the main paper, and all our experimental results for GATA have this correction implemented.

### H.5 Training Curves

Here we present accompanying training curves for experiments reported in this work. We report averaged curves of the normalized accumulated reward with bands representing the standard deviation.

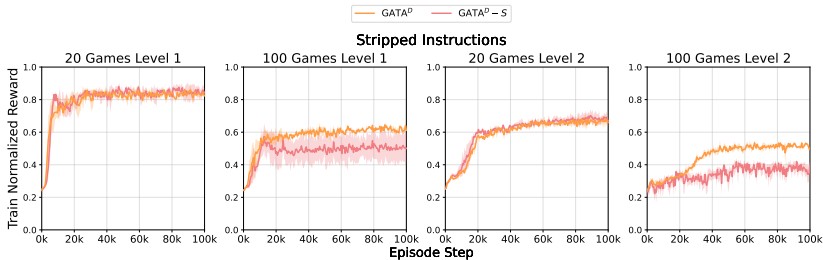

Figure 8: Training curves (of normalized accumulated reward) for the comparison of GATA when trained with instructions (GATA$^D$) versus when instructions are stripped from environment observations (GATA$^D$-S). Agents were trained with 20 or 100 games, at increasing levels of task difficulty (level 1 vs level 2). Bands represent the standard deviation.

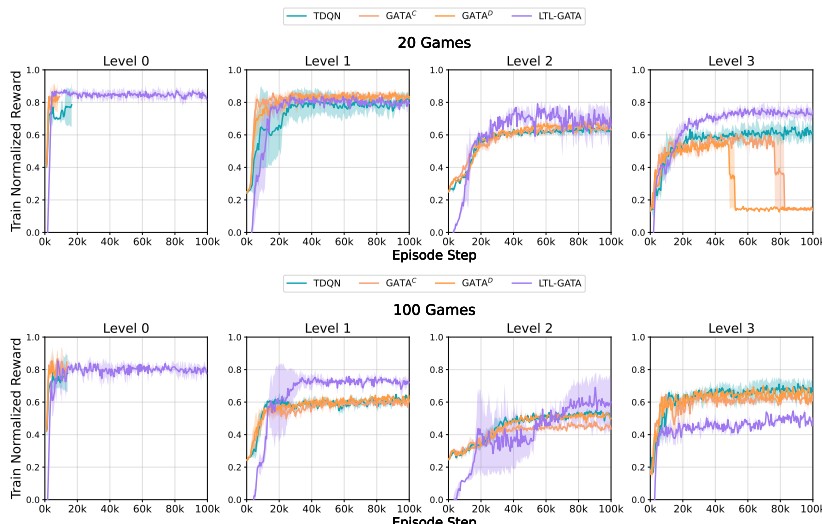

Figure 9: Training curves of the normalized accumulated reward across various levels and on both the 20 (top) and 100 (bottom) game training sets. Bands represent the standard deviation. Note that on level 0, training curves for TDQN, GATA$^C$, and GATA$^D$ were early stopped for achieving $\geq 0.95$ normalized accumulated reward on the validation set for $5$ episodes in a row.

---

[2] https://github.com/xingdi-eric-yuan/GATA-public/blob/c1afc3c9ab38256f839b3e0ddf8243796df5bd77/agent.py#L1353-L1369
[3] https://github.com/xingdi-eric-yuan/GATA-public/blob/c1afc3c9ab38256f839b3e0ddf8243796df5bd77/dqn_memory_priortized_replay_buffer.py#L93-L102

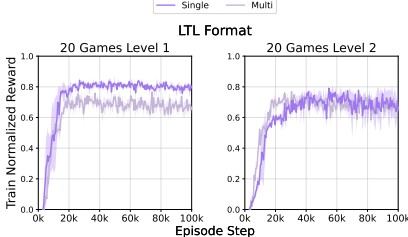

Figure 10: Training curves (of normalized accumulated reward) for the study on LTL predicate format with single-token (*Single*) and multi-token (*Multi*) predicates. Bands represent the standard deviation.

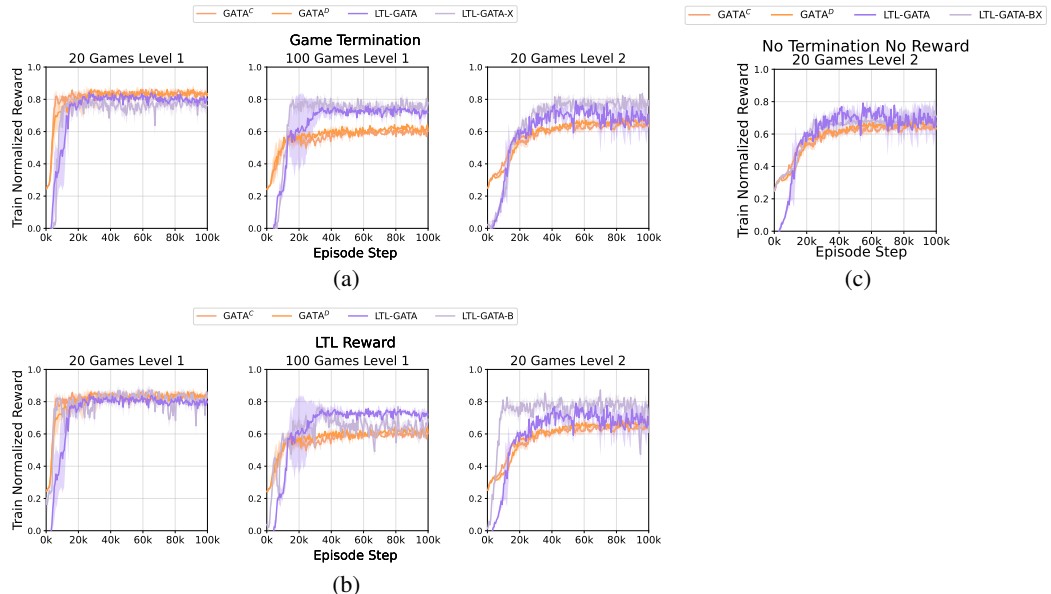

Figure 11: Training curves (of normalized accumulated reward) for the ablation studies on LTL-based episode termination and new reward function $R_{\mathrm{LTL}}(s, a, \varphi)$. (a) LTL-GATA with new reward function $R_{\mathrm{LTL}}(s, a, \varphi)$ and without LTL-based episode termination (*LTL-GATA-X*). (b) LTL-GATA with base game reward function $R(s, a)$ and with LTL-based episode termination (*LTL-GATA-B*). (c) LTL-GATA with base game reward function $R(s, a)$ and without LTL-based episode termination (*LTL-GATA-BX*). Bands represent the standard deviation.

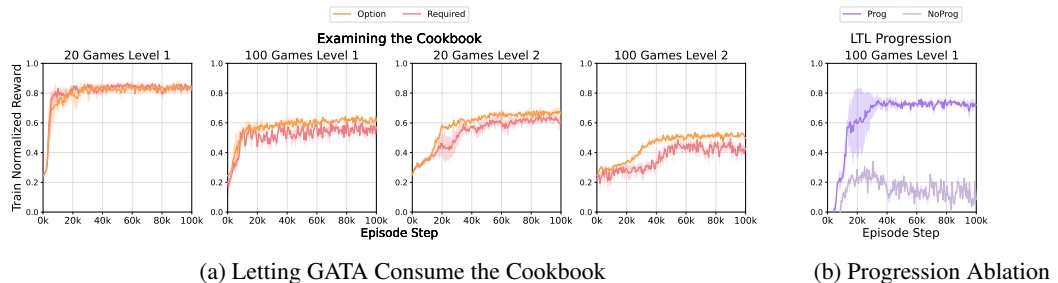

(a) Letting GATA Consume the Cookbook        (b) Progression Ablation

Figure 12: Training normalize reward curves for (a) comparison of GATA$^{\mathrm{D}}$ when given the *Option* to examine the cookbook vs. when it is *Required* to examine the cookbook and (b) comparison of LTL-GATA with (*Prog*) and without (*NoProg*) using LTL progression. Bands represent the standard deviation.

## H.6 Automated Translation: Natural Language Instructions to LTL Details

For the GPT-3 experiments on automated LTL translation in section 5.5, we simply extracted the observations used by our simple translator and saved the observation-translation pairs. Details of that translator were described in in Appendix E and examples of these pairs can be found in Table 3 and Table 4. These observations are the sort of natural language we wish to translate into LTL, and we used six of the observation-translation pairs as the examples in our prompt to GPT-3.

The full prompt to GPT-3 is shown below (with colors added for readability). The six examples consist of a natural language observation (in turquoise) and a corresponding LTL formula (in red) — these remain fixed for all prompts. The seventh line begins with the natural language observation to be translated (in blue).

```
1.  NL: you open the copy of ''cooking :  a modern approach ( 3rd ed .  )''
and start reading :  recipe # 1 ----- gather all following ingredients and
follow the directions to prepare this tasty meal .  ingredients :  cilantro
directions :  dice the cilantro prepare meal  LTL: ('and', ('eventually',
'cilantro_in_player'), ('and', ('eventually', 'cilantro_is_diced'),
('eventually', 'meal_in_player')))

2.  NL: you open the copy of ''cooking :  a modern approach ( 3rd ed .  )''
and start reading :  recipe # 1 ----- gather all following ingredients and
follow the directions to prepare this tasty meal .  ingredients :  pork
chop directions :  chop the pork chop fry the pork chop prepare meal
LTL: ('and', ('eventually', 'pork_chop_in_player'), ('and', ('eventually',
'pork_chop_is_chopped'), ('and', ('eventually', 'pork_chop_is_fried'),
('eventually', 'meal_in_player'))))

3.  NL: you open the copy of ''cooking :  a modern approach ( 3rd ed .  )''
and start reading :  recipe # 1 ----- gather all following ingredients
and follow the directions to prepare this tasty meal .  ingredients
:  black pepper directions :  prepare meal  LTL: ('and', ('eventually',
'black_pepper_in_player'), ('eventually', 'meal_in_player'))

4.  NL: you open the copy of ''cooking :  a modern approach ( 3rd ed
.  )''  and start reading :  recipe # 1 ----- gather all following
ingredients and follow the directions to prepare this tasty meal .
ingredients :  purple potato red onion salt directions :  dice the purple
potato roast the purple potato dice the red onion fry the red onion
prepare meal  LTL: ('and', ('eventually', 'purple_potato_in_player'),
('and', ('eventually', 'red_onion_in_player'), ('and', ('eventually',
'salt_in_player'), ('and', ('eventually', 'purple_potato_is_diced'),
('and', ('eventually', 'purple_potato_is_roasted'), ('and', ('eventually',
'red_onion_is_diced'), ('and', ('eventually', 'red_onion_is_fried'),
('eventually', 'meal_in_player'))))))))

5.  NL: you open the copy of ''cooking :  a modern approach ( 3rd ed .  )''
and start reading :  recipe # 1 ----- gather all following ingredients and
follow the directions to prepare this tasty meal .  ingredients :  black
pepper parsley salt directions :  dice the parsley prepare meal  LTL:
('and', ('eventually', 'black_pepper_in_player'), ('and', ('eventually',
'parsley_in_player'), ('and', ('eventually', 'salt_in_player'), ('and',
('eventually', 'parsley_is_diced'), ('eventually', 'meal_in_player')))))

6.  NL: you open the copy of ''cooking :  a modern approach ( 3rd ed
.  )''  and start reading :  recipe # 1 ----- gather all following
ingredients and follow the directions to prepare this tasty meal
.  ingredients :  purple potato white onion yellow bell pepper
directions :  roast the purple potato roast the white onion dice
the yellow bell pepper prepare meal  LTL: ('and', ('eventually',
'purple_potato_in_player'), ('and', ('eventually', 'white_onion_in_player'),
('and', ('eventually', 'yellow_bell_pepper_in_player'),
('and', ('eventually', 'purple_potato_is_roasted'), ('and',
```

```
('eventually', 'white_onion_is_roasted'), ('and', ('eventually',
'yellow_bell_pepper_is_diced'), ('eventually', 'meal_in_player')))))))
```

7.  NL: you open the copy of "cooking :  a modern approach ( 3rd ed .  )"
and start reading :  recipe # 1 ----- gather all following ingredients and
follow the directions to prepare this tasty meal .  ingredients :  banana
red hot pepper yellow potato directions :  chop the banana fry the banana
chop the red hot pepper fry the red hot pepper slice the yellow potato fry
the yellow potato prepare meal  LTL:

In Table 7, we show how *Da Vinci* and *Ada* complete the prompt to translate this example of
a natural language TextWorld observation. While *Da Vinci* produces the correct LTL transla-
tion, *Ada* struggles and hallucinates incorrect predicates such as 'zucchini_fry_player' and
'banana_is_frozen'.

Table 7: GPT-3 Natural-Language-to-LTL Translation Example

| **Observation** |
|---|
| you open the copy of "cooking :  a modern approach ( 3rd ed .  )"  and start reading :  recipe # 1 ----- gather all following ingredients and follow the directions to prepare this tasty meal .  ingredients :  banana red hot pepper yellow potato directions :  chop the banana fry the banana chop the red hot pepper fry the red hot pepper slice the yellow potato fry the yellow potato prepare meal |
| ***Da Vinci* LTL Translation (*Absolutely Correct*)** |
| ('and', ('eventually', 'banana_in_player'), ('and', ('eventually', 'red_hot_pepper_in_player'), ('and', ('eventually', 'yellow_potato_in_player'), ('and', ('eventually', 'banana_is_chopped'), ('and', ('eventually', 'banana_is_fried'), ('and', ('eventually', 'red_hot_pepper_is_chopped'), ('and', ('eventually', 'red_hot_pepper_is_fried'), ('and', ('eventually', 'yellow_potato_is_sliced'), ('and', ('eventually', 'yellow_potato_is_fried'), ('eventually', 'meal_in_player'))))))))))) |
| ***Ada* LTL Translation (*Incorrect*)** |
| ('and', ('eventually', 'banana_in_player'), ('and', ('eventually', 'red_hot_pepper_in_player'), ('and', ('eventually', 'yellow_potato_in_player'), ('and', ('eventually', 'zucchini_fry_player'), ('and', ('eventually', 'banana_is_frozen'), ('eventually', 'meal_in_player')))) |

# I   Broader Impact

As Adhikari et al. (2020) suggested, text-based games can be a proxy for studying human-machine interaction through language. Human-machine interaction and relevant systems have many potential ethical, social, and safety concerns. Providing inaccurate policies or information or partially completing tasks in critical systems can have devastating consequences. For example, in health care, improper treatment can be fatal, or in travel planning, poor interactions can lose a client money.

Adhikari et al. (2020, section 7) identified several research objectives relating to language-based agents: improve the ability to make better decisions, allow for constraining decisions for safety purposes, and improve interpretability. We highlight how RL agents equipped with LTL instructions can improve in these areas. For constraining decisions, it may be desirable to do so in way that depends on the history, which LTL gives a way to keep track of. With respect to interpretability, we propose that monitoring the progression of instructions provides a mechanism for understanding where and when an agent might be making incorrect decisions, and provides the opportunity to revise instructions or attempt to fix the problem by other means.

On the other hand, instruction following, especially when instructions are not completely specified, may not always be beneficial and can even be harmful. Ammanabrolu et al. (2022) describe a good example where an agent in the Zork1 game breaks into a home and steals the items it needs. In that specific case, breaking into the home has no adverse effect on the agent's reward, so the agent has no incentive not to perform this act. Violation of social norms like this are not modelled in our work, and can have negative impacts, even in less extreme cases. Furthermore, there are potential dangers of incorrect, immoral, or even misinterpreted instructions that lead to dangerous outcomes. Although we do not directly address these concerns in this work, they pose interesting directions for future work.