# OpenReview forum: "Learning to Follow Instructions in Text-Based Games"
_NeurIPS.cc/2022/Conference — NeurIPS 2022 Accept_

### Official Review · Reviewer_CeYJ · 2022-07-02

**Rating:** 5
**Confidence:** 4
**Soundness:** 3 good
**Presentation:** 2 fair
**Contribution:** 2 fair

**Summary:**

The paper proposes to incorporate Linear Temporal Logic (LTL) into a TextWorld game agent (GATA) to (1) provide augmented rewards based on subgoal completion, (2) encode text instruction. Authors find that GATA cannot properly attend to task instructions, and the performance does not drop even if instruction information is not given. Instead, LTL-GATA improves test results in Level 1-2 of TextWorld, but still limited in Level 3 by navigation.

**Questions:**

- In GATA, is a text instruction of the whole task always fed to the model (as part of text observation)? It seems so in Figure 2, just want to confirm.
  - If so, is there any hypothesis why?
  - Similarly, why GATA fails to attend to cookbook information?
  - Is cookbook information same as text instruction of the whole task?
- Line 37: Why "partial completion of tasks can be unsafe"?
- Between line 179-180: what is $\Phi$ in $R_{\Phi}$?


**Limitations:**

The limitation about navigation is briefly discussed, but I think more limitations should be discussed (see weaknesses).

**Strengths And Weaknesses:**

**Strengths**:
- The analysis of LTL failing to attend to instructions points to important problems in TextWorld agent design.
- The introduction of LTL is novel in text games.

**Weaknesses**:
- The title is a bit misleading. Text-based games can mean synthetic games like TextWorld or harder human-designed games like Zork I, but apparently the method cannot apply to the latter. Maybe better to explicitly say "TextWorld" instead of "Text-based Games".

- The formula of LTL seems very simple in TextWorld, and I'm not sure if the approach can be useful beyond simple, synthetic domains with clean logic. Experiments on more complex or fuzzy (e.g. more natural language) domains will make the work much stronger.

- I feel the comparison of LTL-GATA and GATA is a bit unfair, as the former is given with more processed information (e.g. turning NL instructions into LTL synthetic language, and subtask completion maintenance) which assumes ground truth access to the "logic" of the instruction. It'd be better if the model can learn to turn raw game observations into the logic itself, instead of relying on gold annotations.

- The introduction of LTL and LTL progression is formal but not very intuitive. Maybe combine some examples from Section 4.1 is better.

---

> ### Author Response · Authors · 2022-08-02
> **Response to Review of Paper12170 by Reviewer CeYJ**
>
> Thanks for the constructive review. We agree that the study performed in this paper, showcasing the lack of instruction following in SOTA (GATA) TextWorld agents, is important, and that the novel application of LTL to augment GATA for this purpose presents a novel contribution to text-based games.
>
> Regarding the title, we believe the use of “Text-Based Games” (vs restricting to “TextWorld”) is indeed warranted. While the implementation and experiments described in this paper were restricted to TextWorld, the general approach of using LTL as an internal representation for natural language instructions and the exploitation of LTL’s compositional syntax and semantics to monitor progress towards instruction completion are not limited to TextWorld. However, to your point, this is not supported by experimental evidence within the paper. We're confident that we can make this clear in the text. Thanks for raising the issue.
>
> Please see response to Reviewer HDhj “[Q1: Diversity of LTL Formulas]” for general comments regarding the nature of instructions in TextWorld.
>
> In comparing GATA and LTL-GATA, you are correct that LTL-GATA has some information that is more processed (the LTL formulas representing instructions). The comparison was intended to assess overall performance. One way to make the comparison more “fair” would be to further automate the translation of natural language into LTL instructions; see the response to Reviewer QS9K for more on translating natural language to LTL. Note that even in the current version of LTL-GATA, the “subtask completion maintenance” (progression) can be done in a fully automated way given the LTL formula and the GATA belief state. Importantly, progression as we use it does not have access to the ground truth of which propositions hold, but just to what is believed according to the GATA belief state (line 175).
>
>
> Response to Reviewer Questions:
>
> **Q1:** GATA does not get the whole text instruction on every time step. Figure 2 shows a step in which the cookbook is read, and that is why the recipe is observed in that step. The agent might never read the cookbook in an episode – indeed, as we describe in section 3, using GATA the agent usually does not look at the cookbook in levels 1 and 2. In those cases it won’t ever observe the recipe.
>
> As to why GATA fails to attend to the cookbook information in those cases in which it has received it, a hypothesis may be suggested by our progression ablation experiment (Figure 5). In that experiment, we considered a version of LTL-GATA in which the LTL formula is never progressed, so after the agent examines the cookbook, it receives the same LTL encoding of the entire recipe on each subsequent timestep. (Note again that the textual observation of the recipe only occurs at the point the cookbook is examined.) This version of LTL-GATA, lacking progression to keep track of what parts of the recipe had already been completed, performed poorly. So perhaps GATA’s failure to attend to the cookbook information is caused by the lack of a progression-like mechanism to keep track of what should be done when.
>
> Technically, following the recipe text is not quite the whole task: the agent also needs to eat the meal afterwards (however, that part is always the same and is appended in our LTL translation of the recipe, as described in Appendix E).
>
> **Q2:** Why partial completion of tasks can be unsafe is elaborated upon in Appendix I. The Cooking domain is not realistic enough to include safety concerns, but in the real world, partially completing a task may leave the world in an unsafe state. For example, helping someone half-way across a street leaves them in a position vulnerable to traffic. In real-world cooking, failing to cook the meat in a recipe may create a health hazard.
>
> **Q3:** The Phi in R_Phi is just a decoration to distinguish it from R; we could have named R_Phi as R’ instead.

---

> > ### Comment · Reviewer_CeYJ · 2022-08-05
> > **Thank you!**
> >
> > Thanks authors for the response, which addressed some of my questions. Only considering main points:
> > - I still don’t know how this idea could easily extend to more complex domains like Jericho, but authors also acknowledge current draft does not have experiment results to support this.
> > - Authors also acknowledge LTL currently requires extra supervision, but reading other reviews I partly agree in the LLM era this might not be too much of an issue (esp. given simple domains like TextWorld).
> >
> > Given these, I want to raise my score from 4 to 5.

---

### Official Review · Reviewer_QS9K · 2022-07-02

**Rating:** 8
**Confidence:** 5
**Soundness:** 3 good
**Presentation:** 4 excellent
**Contribution:** 3 good

**Summary:**

In this work, the authors first identity some important issues in recent text-based game agents that they tend to ignore instructions, these instructions can be dominant in solving text-based games. Then, the authors propose to equip agents with an internal structured representation of instructions, in the form of LTL, a formal language.

Designed for the cooking games in TextWorld, the authors use three types of instructions:
1. the agent needs to examine the cookbook to know the goal of the current game (which changes in every game);
2. the information written in the cookbook;
3. go to the kitchen if the agent is spawned elsewhere.

The instructions can be used in multiple ways:
1. (main use) as an additional input being fed into the model, it can guide the agent to act in a favorable manner.
2. (additional use) to help shaping rewards, but as the authors discuss in Appendix H.4.2, this is not very helpful.

The proposed agent contains a minimal modification from prior work (the LPL updater), the authors show that equipped with the LPL module, their agent clearly outperforms baselines (GATA).




**Questions:**

1. Please see weakness 1.
2. Related to Q1, it would be nice if the authors could discuss some potentials using large pre-trained language models (e.g., GPT-3) for instruction generation, then using similar large LMs (e.g., Codex) to translate natural language instructions into formal language (e.g., LTL).
3. Regarding the first implementation issue discussed by the authors [HERE](https://github.com/xingdi-eric-yuan/GATA-public/blob/c1afc3c9ab38256f839b3e0ddf8243796df5bd77/dqn_memory_priortized_replay_buffer.py#L120-L123) and authors' code (experience_replay.py, L122-L129). I agree that length should be `n`, but the reward line should have `self._buffer[idx + j + i + 1]` instead of `self._buffer[idx + j + i]`? This is because that at a game step $t$, what's being pushed into the replay buffer is the reward $r_{t-1}$ resulted by the action at previous game step. Correct me if I'm wrong.



**Limitations:**

The authors discussed broader impact including potential societal concerns in the appendix. I agree and I do not see major issue regarding this.

**Strengths And Weaknesses:**

- Strengths
1. The authors identify issues in existing (top performing) RL agents, and aim to mitigate these issues. The motivation is clear.
2. The authors show results that equipped with the proposed LPL module, GATA can obtain a performance gain on all the settings, this is especially obvious in level 2 games, where the tasks are the most difficult (ignoring navigation). The contribution is solid.
2. The proposed system contains only minimal modification from prior work's model architecture, it is easy to compare the proposed method against baselines.
3. The presentation if clear, the idea is easy to follow.
4. Using success rate + game score as metric makes a lot of sense.
5. The authors point out some implementation issues of prior works and provide bug fixes.

- Weaknesses
1. The authors fails to sufficiently discuss the weaknesses of LPL-aided agents. For instance, it seems that for every domain (e.g., cooking, crafting, navigation), the proposed system requires to some level human annotated instructions. To what extend do the proposed instructions scale or transfer? Given a new domain (either a simulation environment or some real world tasks), what are ways to design good instructions? I am not against leveraging humans as domain experts, but these discussion can help the readers to better understand the bigger picture.
2. I suggest the authors to include numbers in the appendix, so future works can compare with their numbers.

- Typo
1. L192, Table 3 --> Figure 3

---

> ### Author Response · Authors · 2022-08-02
> **Response to Review of Paper12170 by Reviewer QS9K**
>
> Thanks for your detailed technical review and suggestions. We respond to your questions below.
>
> **[Q1: LTL instructions]** In our implementation, we had a handcrafted script to convert natural language instructions in the Cooking domain into LTL. Automated translation of natural language into LTL was outside the scope of our work, but there is research in that area, and we believe that such automated translation could make it easier to apply our approach to new domains (see response to Q2 below).
>
> We also note that just applying standard RL techniques to a game or real-world task requires designing a reward function, which can be difficult and in some cases may require similar expertise to determining what sorts of LTL instructions should be followed.
>
>
> **[Q2: Language models and translating natural language into LTL]** Thanks for bringing up this exciting research direction. We agree that there is potential in using large language models (LLMs) to translate natural language instructions into LTL. There is a history of research on automated translation of natural language into LTL and other temporal logics from the pre-LLM era [e.g., Dzifcak et al., 2009; Finucane et al., 2010], and recent work using language models [e.g., Hahn et al., 2022].
>
> Additionally, there has been work using LLMs to generate plans for completing tasks, given a high-level description of the task [Huang et al., 2022]. This suggests the possibility for text-based games of using an LLM to generate detailed natural language instructions beyond those explicitly given in the game, instructions which could then also possibly be converted into LTL.
>
> We will add this discussion to the paper.
>
>
> **[Q3: The replay buffer in the code]** The reason we have self._buffer[idx + j + i] instead of self._buffer[idx + j + i + 1] is because what is being pushed into the replay buffer is the reward r_t resulting from the action at the current game step, not r_t-1 from the previous step. We can see this in train.py:
>
> * Line 317 calculates the step-based reward, giving the current reward for the current action
> * Line 332 stores that in game_rewards
> * Line 407, we pushed the relevant reward and action to the buffer
>
>
> # References
>
>
> [Dzifcak et al., 2009] Dzifcak, J., Scheutz, M., Baral, C., and Schermerhorn, P.. What to do and how to do it: Translating natural language directives into temporal and dynamic logic representation for goal management and action execution. 2009 IEEE International Conference on Robotics and Automation, May 2009, pp. 4163–4168.
>
> [Finucane et al., 2010] Finucane, C, Jing, G, and Kress-Gazit, H. LTLMoP: Experimenting with language, Temporal Logic and robot control. 2010 IEEE/RSJ International Conference on Intelligent Robots and Systems, Oct 2010, pp. 1988–1993.
>
> [Hahn et al., 2022] Hahn, C., Schmitt, F., Tillman, J. J., Metzger, N., Siber, J., and Finkbeiner, B. Formal Specifications from Natural Language. arXiv preprint arXiv:2206.01962, 2022.
>
> [Huang et al., 2022] Huang, W., Abbeel, P., Pathak, D., Mordatch, I. Language Models as Zero-Shot Planners: Extracting Actionable Knowledge for Embodied Agents. arXiv preprint arXiv:2201.07207, 2022.

---

> > ### Comment · Reviewer_QS9K · 2022-08-04
> > **My decision to support this work does not change**
> >
> > Thanks for the response. I have read all the reviews and comments.
> >
> > The authors have replied and addressed well my concerns and questions. My decision to support this work does not change.
> >
> > ### Message to co-reviewers
> >
> > 1. In my opinion, the direction of leveraging formal languages (including LTL and many others) in solving text-based games (including TextWorld and human-made text adventure games, as mentioned by Reviewer CeYJ), or even more broadly, interactive reasoning tasks, is quite neat and can have great potentials. I see this work as a nice starting point:
> >
> > * the authors empirically show current SOTA agent has crucial shortcomings (overfitting on averaged action sequences without reading the recipes which are changed in every episode);
> > * they make minimal change on GATA and show how and why LTL can help.
> > * they propose a useful evaluation metric.
> >
> > Therefore, I believe the work contains adequate merits.
> >
> > 2. I understand and agree with other reviewers on the generalizability concern (see weakness 1 in my review). However, as the authors mentioned in their response, many recent work have started to explore how to leverage foundation models (e.g., pre-trained LLMs) in interactive environments including embodied worlds and text-based worlds [1][2][3][4]. Recent work show that if prompted properly:
> >
> > * a. LLMs can be used as a queryable knowledge source, which can to some extent return both declarative knowledge (e.g., peppers are edible) and procedural knowledge (e.g., to cook lasagna, one needs to first ...);
> > * b. LLMs can be used as an agent's memory so they keep track of their experience so far.
> >
> > I see some rather clear connection between the above points with automatic instruction generation (maybe in formal language). That said, foundation models could potentially alleviate the pain of domain gaps. If an agent is mastering a set of cooking tasks (aided by LTL), it may be able to adapt to crafting tasks given some instructions generated by an LLM (probably in a 0-shot manner).
> >
> > I am quite excited and I really look forward seeing improvement in this sub-area. The hand-crafted domain knowledge used in this work is now less concerning me.
> >
> > 3. Re my Q3 regarding replay buffer code: the authors are right.
> >
> >
> > ### References
> > * [1] Do As I Can, Not As I Say: Grounding Language in Robotic Affordances. Ahn et al., 2022. [link](https://arxiv.org/abs/2204.01691)
> > * [2] Inner Monologue: Embodied Reasoning through Planning with Language Models. Huang et al., 2022. [link](https://arxiv.org/abs/2207.05608)
> > * [3] Pre-train, Prompt, and Predict: A Systematic Survey of Prompting Methods in Natural Language Processing. Liu et al., 2021. [link](https://arxiv.org/abs/2107.13586)
> > * [4] Socratic Models: Composing Zero-Shot Multimodal Reasoning with Language. Zeng et al., 2022. [link](https://arxiv.org/abs/2204.00598)
> >
> > I am happy to discuss any of the above further.

---

### Official Review · Reviewer_HDhj · 2022-07-12

**Rating:** 5
**Confidence:** 4
**Soundness:** 3 good
**Presentation:** 4 excellent
**Contribution:** 2 fair

**Summary:**

This paper shows that the performance of the GATA agent is not affected by stripping the text instructions from the environment observations. This shows that the agent doesn’t consider the task instruction when making decisions. This paper proposes a method to improve task completion by exploiting progress monitoring in LTL. The resulting agent shows improvements in the Cooking domain at level 1 and 2.

**Questions:**

- What’s the diversity of the converted LTL formulas? From the paper and the supplemental materials, the instructions are mainly concatenated with Eventually and Next. If the instructions are in simple forms, we may only need to co-train an event detector to match the events mentioned in the instructions without LTL formulas. Ablation of the LTL formula but keeping the event detector can show how much the text-based games require LTL understanding.
- The experiments show most improvements in level 1 and 2, but not 3 because navigation is still an issue. Is it possible that the progress of navigation can be encoded as the LTL formulas so the same method can help navigation as well?

**Limitations:**

While a simple LTL translator works for the Cooking domain, for more complex text-based instructions, the efficacy of the proposed method is limited by the quality of the semantic parser that parses natural language sentences into LTL formulas. It would be useful if the authors can discuss the impact of parser quality.

**Strengths And Weaknesses:**

Strength
- This paper shows an analysis that reveals the weakness of the previously proposed model.
- The experiments at different levels and ablations show how incorporating LTL in text-based games can help task execution.

Weakness
- Novelty: Exploiting LTL progress monitoring to improve RL agents’ performance has been demonstrated in non-text-based games [Vaezipoor 2021]. It is good that the same method is applied to the text-based games but this shows limited novelty of the proposed method. It will be helpful if the authors can discuss how LTL can affect the training or design of the agents in text-based games compared to non-text-based games.
- Generalization: GATA was evaluated in 500+ games in TextWorld. However, the proposed method is only evaluated in the Cooking domain. It is unclear how the model can be extended to other domains.

---

> ### Author Response · Authors · 2022-08-02
> **Response to Review of Paper12170 by Reviewer HDhj**
>
> Thanks for your thorough review and for the good questions.
>
> Two clarifications, followed by answers to explicit questions.
>
> [Novelty:] Indeed, LTL has begun to show great promise in RL as recently demonstrated by Vaezipoor et al. (2021) and other works. The architecture proposed here, the first to our knowledge to propose formal language exploitation within a text-based game setting, builds on this work with several important differences. Those works are typically deployed in fully observable settings, and assume the existence of so-called “event detectors” that establish ground-truth propositional values (e.g. “carrot is chopped”, “knife is held”) at both training and deployment time. This supports “perfect” evaluation of LTL formulae. In a text-based game setting there are no event detectors, and the environment is partially observable so GATA must learn a belief graph to establish these propositional values, to evaluate satisfaction of LTL instruction steps, and to progress instructions. (This is in contrast to the standard fully-observable exploitation of LTL in RL.) Interestingly, the structured relational representation afforded by the GATA belief graph supports generalization. Indeed, our experiments demonstrated generalization to unseen recipes in the Cooking domain, including novel combinations of objects and relations/verbs. Successful application of LTL in partially observable environments with imperfect propositional state estimation is an important milestone for this area of RL research. This is an important question and we will elaborate in the paper. Thanks for raising it.
>
> [Generalization:] By design, our evaluation followed the specs of the original GATA evaluation, including the full test setup. Like GATA, we restricted our attention to the TextWorld Cooking domain. Like GATA, we used 100+ “games” at different levels. The “500+ games” that GATA evaluates on come from levels 1-4 in the Cooking domain. Each level has 140 unique games, resulting in 560 (140 * 4) unique games. To have as fair a comparison with GATA as possible, we re-used the identical set of games together with their data loading code (which we thank them for). Our levels 0, 1, 2, & 3 are from this provided set of GATA games (https://github.com/xingdi-eric-yuan/GATA-public/blob/c1afc3c9ab38256f839b3e0ddf8243796df5bd77/README.md?plain=1#L88). One difference to note in our assessment is that “Level 4” from the reported GATA work is omitted, replaced by “Level 0”. GATA’s Level 4 is an augmentation of Level 3 that adds more ingredients, and both GATA and LTL-GATA at this level suffer from the navigation issues we reported in our paper. As we wanted to focus on instruction following and not navigation, we omitted this level and chose to use Level 0 instead.
>
> [Q1: Diversity of LTL Formulas] Thanks – an important question. The instructions found in the TextWorld domains that form the basis of the GATA work and our analysis only required a subset of LTL: conjunctive formulae involving the Next and Eventually temporal modalities. As observed by the reviewer, monitoring progress towards satisfaction of such simple-form instructions could be realized via a simple mechanism without appealing to the compositional syntax and semantics of LTL. However, such an approach fails very quickly when instructions become even a little more complex. Consider the simple instruction to “fry, boil, or bake the carrots and then fry the onions,” a simple mechanism might force the agent to try to fry AND boil AND bake the carrots, rather than realizing that once one of the disjuncts was satisfied that the instruction step was completed. Similarly, adding the simple instruction step “keep the carrots warm” necessitates the Always and/or the Release temporal modalities of LTL. Monitoring satisfaction of such natural language instructions is trivially handled by our approach which exploits the rich temporal expressiveness of LTL together with its compositional syntax and semantics.
>
> [Q2: “Is it possible that the progress of navigation can be encoded as the LTL formulas so the same method can help navigation as well?”]  Definitely! Exploring at test time to find items and rooms in an unknown environment is a major challenge built into many text-based games, as we see in the TextWorld Cooking domain. Optimal exploration generally requires memory to remember where the agent has been. LTL can be used to dictate strategy and/or simply to track such exploration. E.g., if the agent is looking for something, it’s useful to remember which rooms it has previously checked. LTL can also be used to encode (learned) navigation instructions (e.g. “find the blue door, go through it and then turn right”). Note that such information is not provided by TextWorld, and this vector of research is beyond the scope of this paper, but we were excited to respond because the reviewer’s instincts speak to the diverse opportunities afforded by leveraging the work presented here.

---

> > ### Comment · Reviewer_HDhj · 2022-08-06
> > **Thanks for addressing my questions!**
> >
> > I would like to thank the authors for addressing my questions. I agree and am excited about the potential to incorporate formal semantics in interactive reasoning tasks.
> >
> > - Novelty: Thanks for the clarification. Using belief graphs for progress monitoring instead of an event detector is an important difference compared to prior work in fully observable environments. It will be helpful to incorporate discussion in the final paper.
> >
> > - Generalization: My original question was intended to ask how to generalize to other games or more complex text game interactions. I can see that using LLMs can partly address the issue of instruction/formula generation (e.g. simple predicates or procedures) but it is still unclear how this approach can easily extend to more complex games and how auto-generation keeps complex temporal logic semantics.
> >
> > I will raise my score from 4 to 5.

---

### Official Review · Reviewer_erG7 · 2022-07-12

**Rating:** 4
**Confidence:** 3
**Soundness:** 3 good
**Presentation:** 2 fair
**Contribution:** 2 fair

**Summary:**

This paper proposes a new agent for Text Based Environment (TBG). The new agent is based on Linear Temporal Logic (LTL) and address the shortcomings of the existing SOTA agent GATA (Graph Aided Transformer Agent). Authors show that GATA agent is agnostic to instructions and are typically not successful in completing tasks in TBG. Authors incorporate formal representation of natural language instructions (via LTL) and this provides the agent with a measure of progress in task completion and helps the agent to complete tasks.

**Questions:**

None

**Limitations:**

The main limitation of the proposed model is that it is not generalizable to many (and possibly unknown) domains. See also limitation above.

**Strengths And Weaknesses:**

Strengths:

1. Authors experimentally show that the SOTA agent GATA largely ignores initial instructions while playing the game. Further, authors show that the agent is ignorant of the final goal of the game (in this the agent does not know what it is suppose to cook as the final goal), the agent is solely driven by rewards alone. This is an interesting empirical insight by the paper.

2. Authors develop and add a new component to the existing GATA architecture to overcome its limitation. The new component is a LTL Updater that generates LTL instructions from natural language observations and it also progresses the instructions.



Weaknesses:

1. The use of formal structured language improves the performance but it is not generalisable to other domains. Authors in this case develop the specifications for cooking domain but if the agent were to be deployed for another domain new set of specifications would be required. LTL limits the possibility of deploying an agent to a real world setting where agent would have to deal with different tasks from varied domains. In such cases it is not possible to come up with specification for each possible domain as the number of such possibilities in principle is infinite.

---

> ### Author Response · Authors · 2022-08-02
> **Response to Review of Paper12170 by Reviewer erG7**
>
> Thanks for your review. We appreciate your point about generalization. In this work, GATA provides the domain-specific vocabulary for describing properties of state, and our LTL augmentation provides the *domain–independent* temporal modalities (Next, Eventually, etc.) and logical connectives for composing those properties of state into instructions. In this way the proposed LTL technique is very general, limited only by GATA’s ability to recognize properties of state.
>
> Nevertheless, to your point, in our current implementation we used a domain-specific script to translate natural language into LTL. The domain-specific vocabulary was provided by GATA, so the set of possible formulas for a domain was not arbitrarily open-ended. Please see also our comments to Reviewer QS9K regarding the automated translation of natural language into LTL, which is an active area of research that we believe is promising (but outside the scope of what we were tackling in this paper).
>
> We agree that generalization is an important point, and we’ll add further discussion on that to the paper.

---

### Author Response · Authors · 2022-08-02
**General Comments for All**

Thank you to all the reviewers for the substantive feedback and thoughtful comments. We are confident that we can address all feedback within a minor edit/revision of the paper.

Reviewers recognized the importance of the experimental analysis undertaken in this work to study instruction following by text-based game RL agents – a topic that has heretofore not been analyzed in depth. Experiments showed that when instructions were eliminated altogether, the SOTA TextWorld agent (GATA) showed no change in performance, suggesting that it was not exploiting the instructions. Further, when GATA had the instructions, they were not fully executed in a reasonably large number of cases – something that could be unsafe in non-game settings, and that was not evident via standard GATA evaluation metrics. Reviewer erG7 found this “an interesting empirical insight”. Reviewers HDhj and CeYJ specifically praised the analysis as a strength. Reviewer QS9K approved of the new proposed metric (success_rate + game score).

To further study and address the task of instruction following, the paper proposed equipping RL agents with an internal structured representation of natural language instructions as Linear Temporal Logic (LTL). The resulting text-based game agent was able to exploit the compositional syntax and semantics of LTL to advance and monitor progress towards instruction completion. Reviewer CeYJ noted the novelty of the proposed method, and reviewer QS9K appreciated how the approach, by performing a “minimal modification” to the baseline GATA architecture, “clearly outperforms [the] baselines (GATA)”, and found that the “contribution is solid” (quotes from QS9K).

We address reviewer-specific questions in individual responses to the reviewers.

If we have addressed concerns please consider updating review scores as appropriate.

---

### Meta-Review · Area_Chair_pBoD · 2022-08-26

**Recommendation:** Accept
**Confidence:** Certain

**Metareview:**

This paper proposes a linear temporal logic-based solutions to some of the common failure modes of existing agents addressing text-based games, stemming from the analysis of these failure modes. There was disagreement about whether the work is publishable based on the limited range of domains LTL supports. That said, from reading the discussion, it seems that the emerging consensus is that it is sufficiently novel and interesting for this work to prove the concept in text domains, and that further work may seek to extend such approaches (or draw inspiration from them) in a wider range of domains. I support acceptance.

**Award:**

No

---

### Decision · Program_Chairs · 2022-09-14

Accept